# OTIS: Learning High-Quality Time Series Features With Tiny Encoders

## Abstract

We introduce OTIS, an **o**pen **ti**me **s**eries encoder that yields high-quality time series features for downstream deployment on *any* system, including resource-constrained wearables and industrial sensors.[1] Currently, the development of powerful general-purpose encoders relies on the scaling laws hypothesis, using large encoder sizes to memorise the heterogeneous distributions of multi-domain training data. However, this reliance on scale creates a barrier to real-world utility, rendering deployment on resource-constrained systems infeasible due to strict memory, energy, and latency constraints. Surprisingly, we find that tailoring standard masked modelling pre-training to time series properties yields a tiny 7.1 M encoder that matches the state-of-the-art performance of 54× larger encoders across 162 tasks, while requiring 10× less memory, 43× less energy, and 37× lower latency. To achieve this without the capacity tax, we introduce three novel components: (1) a *domain-aware tokeniser* to resolve conflicting semantics within multi-domain training data; (2) a *dual masking strategy* to capture spatiotemporal structures and temporal causality; and (3) a *structure-aware objective* to decouple feature learning from modelling noise. Consequently, OTIS produces high-quality time series features that enable state-of-the art performance in discriminative tasks and even extend seamlessly to generative tasks at minimal additional cost. To democratise access to powerful time series features on any system, we release our code and pre-trained weights.

## 1 Introduction

Learning hiqh-quality features from unlabelled data has transformed natural language processing and computer vision (CV). Driven by self-supervised learning on large-scale data, the field has evolved from specialised encoders to general-purpose encoders capable of solving a variety of tasks across different domains. In CV, encoders like DINOv3 (Siméoni et al., 2025) learn visual features (shapes, textures, and depth) that transfer seamlessly across tasks (classification, segmentation, and object detection) and domains (natural, satellite, and medical images). Recently, this wave has reached time series analysis, yielding two distinct research streams: (i) *forecasting models* such as Chronos (Ansari et al., 2024) and Time-MoE (Shi et al., 2025) that focus solely on generation, and (ii) *general-purpose encoders* like UniTS (Gao et al., 2024) and MOMENT (Goswami et al., 2024) that serve discriminative but also simple generative purposes.

However, training a general-purpose time series encoder faces a unique challenge: extreme data *heterogeneity*. Time series vary heavily across domains in both structure and semantics. Structural heterogeneity manifests in variate dimensionality, ranging from uni-variate audio to 256-variate electroencephalography (EEG). Semantic heterogeneity emerges from distinct variate correlations: EEG variates exhibit physiological connectivities of the brain, differing fundamentally from the thermodynamic laws governing weather variates. Current approaches, such as the 386 M MOMENT, address this heterogeneity via the *scaling laws hypothesis* (Kaplan et al., 2020), relying on large encoders to memorise data heterogeneity. While effective, this creates a bottleneck for real-world downstream deployment. Time series analysis is predominantly required on resource-constrained systems, from wearable health monitors (Shah et al., 2025) to industrial edge sensors (Wiesmayr et al., 2025), where deploying large-scale encoders is infeasible due to constraints in memory, energy, and latency (Fig. 1).

---

[1]Code and pre-trained weights are available at `https://github.com/OTIS-official/OTIS`.

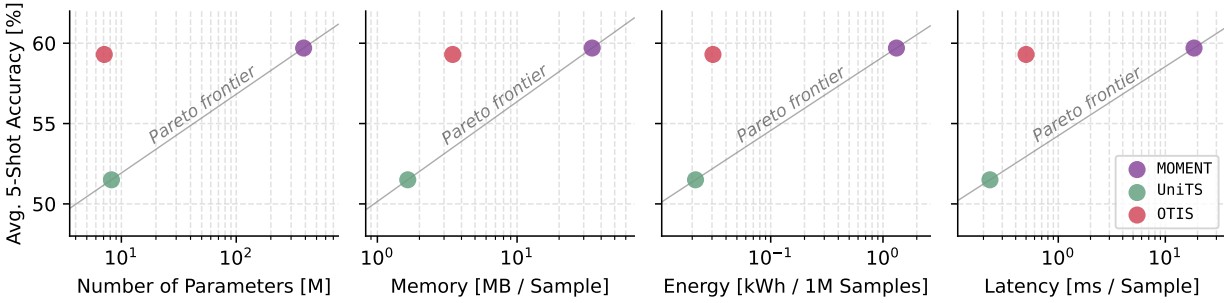

Figure 1: **Extending the Pareto frontier of general-purpose encoders.** Across 149 datasets, `OTIS` matches the state of the art while requiring 54× fewer parameters, 10× less memory, 43× less energy, and 37× lower latency. This marks a large step towards democratising access to high-quality time series features.

In this work, we demonstrate that high-quality time series features can also be learned by tiny encoders via targeted *inductive biases*. Surprisingly, we find that carefully tailoring standard masked data modelling pre-training to time series properties is sufficient to produce `OTIS`, a tiny 7.1 M **o**pen **ti**me **s**eries encoder that matches state-of-the-art performance across tasks and domains previously reserved for large-scale encoders such as the 386 M `MOMENT`. To achieve this with 54× fewer parameters, 10× less memory, 43× less energy, and 37× lower latency (Fig. 1), we introduce three novel components: (1) a **domain-aware tokeniser** (Sec. 3.1) to resolve conflicting semantics within multi-domain training data via learnable domain-specific embeddings; (2) a **dual masking strategy** (Sec. 3.2) to capture spatiotemporal structures and temporal causality via random and post-fix masking, respectively; and (3) a **structure-aware objective** (Sec. 3.3) to decouple feature learning from modelling noise via a normalised cross-correlation loss term. In extensive evaluations on a total of 162 tasks that go far beyond standard benchmarks, we analyse the deployment of general-purpose encoders in real-world scenarios, including epileptic seizure detection from single-channel EEG and cardiac phenotype regression from 12-lead electrocardiography (ECG). These evaluations confirm that `OTIS` provides high-quality time series features for discriminative tasks, which also seamlessly extend to generative tasks at minimal additional cost by reusing the lightweight 1.5 M decoder from pre-training.

## 2 Related Works

**Self-Supervised Learning**   Learning high-quality features from unlabelled data typically relies on carefully designed pretext tasks. To this end, early approaches relied on *contrastive learning*, aligning augmented views of a sample in the feature space (Kiyasseh et al., 2021; Zhang et al., 2022; Woo et al., 2022; Wang et al., 2022; Radhakrishnan et al., 2023; Demirel & Holz, 2023). However, identifying effective time series augmentations is challenging and highly domain-dependent. Consequently, *masked data modelling* (MDM) has emerged as the dominant paradigm. By reconstructing masked segments from a small visible view, methods like `PatchTST` (Nie et al., 2023), `UniTS` (Gao et al., 2024), `MOMENT` (Goswami et al., 2024), and `MOIRAI` (Woo et al., 2024) learn features without hand-crafted augmentations. While effective within single domains, standard MDM struggles in multi-domain settings, where data heterogeneity renders learning of high-quality features difficult.

**General-Purpose Encoders**   The dominant research stream in time series analysis focuses on *forecasting models* such as `GPT4TS` (Zhou et al., 2023), `TTM` (Ekambaram et al., 2024), `MOIRAI` (Woo et al., 2024), `Chronos` (Ansari et al., 2024), `Timer-XL` (Liu et al., 2025), or `Time-MoE` (Shi et al., 2025). These works adapt strategies from natural language processing to reliably predict future horizons from past observations, tailoring feature extraction to serve forecasting. The second stream aims to build *general-purpose encoders* to provide features that serve multiple purposes. For instance, `UniTS` (Gao et al., 2024) uses specialised task prompts to perform discriminative and generative tasks with a single 8.2 M encoder. However, relying on explicit task prompts risks building fixed mappings for each prompt within the encoder rather than learning generalisable features. Other approaches such as the 386 M `MOMENT` (Goswami et al., 2024) substantially outperform their smaller counterparts simply by exploiting the *scaling* laws hypothesis (Kaplan et al., 2020; Hoffmann et al., 2022).

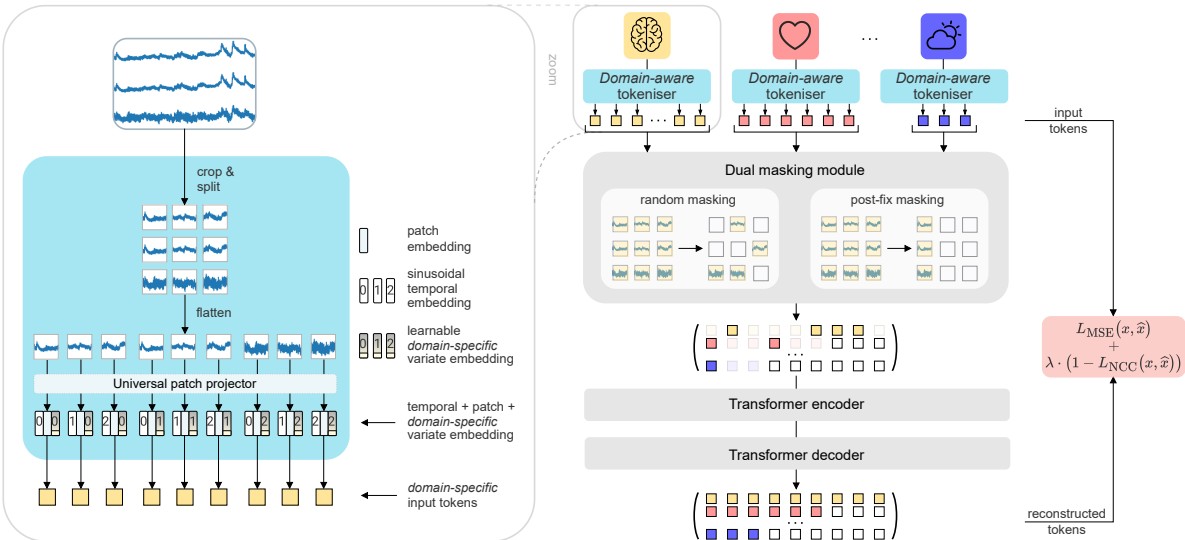

Figure 2: **Effective pre-training on multi-domain data.** Patches are embedded via a universal patch projector. Sinusoidal temporal embeddings and learnable domain-specific variate embeddings encode positions. Random and post-fix masking is applied to the input, which is reconstructed using a structure-aware loss.

However, relying on scale severely limits deployment on resource-constrained systems like wearables and industrial edge sensors due to strict memory, energy, and latency constraints.

## 3 Methodology

In this work, we demonstrate how the extreme heterogeneity of multi-domain training data can be addressed by design rather than scaling, yielding a tiny **o**pen **ti**me **s**eries encoder (`OTIS`) with state-of-the-art capabilities that is deployable on *any* system. To build `OTIS`, we design a simple and robust training recipe based on widely known time series properties. Particularly, we address multi-domain heterogeneity during tokenisation (Sec. 3.1) and increase the inductive bias in standard masked data modelling (MDM) pre-training with a simple masking strategy (Sec. 3.2) and regularisation (Sec. 3.3). Overall, this simple recipe yields a powerful general-purpose encoder that can be easily adapted to any downstream task in any domain (Sec. 3.4).

### 3.1 Domain-Aware Tokenisation

Time series data exhibits extreme structural and semantic heterogeneity across domains. Given a sample $\mathbf{X} \in \mathbb{R}^{V_S \times T}$ from domain $S$, *structural heterogeneity* refers to diverse variate dimensionality ($V_S$), while *semantic heterogeneity* refers to diverse variate correlations (e.g. physiological connectivity versus thermodynamic laws). This semantic diversity makes pre-training on multi-domain data challenging, forcing the encoder to reconcile conflicting dependencies within a unified representation space. While existing methods rely on scaling encoder and data size to mitigate conflicting learning signals, we address them by design to learn powerful time series features with a tiny, highly deployable Transformer (Dosovitskiy et al., 2021) encoder.

To this end, we introduce a hierarchical tokenisation strategy that explicitly separates the low-level projection of universal local patterns from the injection of domain-specific semantics. We first randomly crop or zero-pad $\mathbf{X}$ to a fixed context length $\overline{T}$ and divide it into $T'$ segments of size $P$. Flattening then yields a sequence of $V_S \cdot T'$ raw time series patches $\mathbf{x}_{v,t} \in \mathbb{R}^{1 \times P}$, with $v \in \{1, \dots, V_S\}$ and $t \in \{1, \dots, T'\}$.

**Universal Patch Projection**  We first project the raw patches into a continuous representation space. To this end, we use a patch projector consisting of a 1D convolutional layer with kernel size and stride $P$, followed by layer normalisation and a GELU activation. This yields *patch embeddings* $\mathbf{e}_{v,t}^{\mathcal{P}} = e^{\mathcal{P}}(\mathbf{x}_{v,t}) \in \mathbb{R}^{1 \times D}$, where $D$ denotes the encoder dimension. As the structure within a single patch arises from local, low-level patterns

that occur in any time series regardless of the domain, e.g. valleys, peaks, or saddle points, this projector is kept entirely independent of the source domain, acting as a universal, non-semantic linear projection.

**Domain-Aware Spatiotemporal Encoding** To allow the subsequent time series encoder to build global context and understand the relationships between these isolated patch embeddings, we equip them with spatiotemporal information via decoupled 2D positional priors. First, we inject *universal temporal information* using fixed 1D sinusoidal embeddings $\mathbf{e}_t^{\mathcal{T}} = e^{\mathcal{T}}(t) \in \mathbb{R}^{1 \times D}$ to encode the sequential order $t$. Since the nature of temporal ordering flows consistently across all domains, the temporal embeddings are universally shared. Second, we inject *domain-specific variate information* using learnable embeddings $\mathbf{e}_{S,v}^{\mathcal{V}} = e_S^{\mathcal{V}}(v) \in \mathbb{R}^{1 \times D}$ to encode the variate position $v$. As the inherent meaning of a single variate and the interaction between multiple variates fundamentally change depending on the source domain $S$, the variate embeddings are kept strictly domain-specific. This allows the encoder to learn the unique variate semantics of each distinct domain.

The final input token fed to the encoder is the sum of these three components: $\mathbf{e}_{v,t} = \mathbf{e}_{v,t}^{\mathcal{P}} + \mathbf{e}_t^{\mathcal{T}} + \mathbf{e}_{S,v}^{\mathcal{V}} \in \mathbb{R}^{1 \times D}$, yielding the complete sequence $\mathbf{E} \in \mathbb{R}^{(V_S \cdot T') \times D}$. This hierarchical tokenisation strategy equips the time series patches with the necessary domain-aware spatiotemporal information, enabling the encoder to focus on dense representation learning without suffering from conflicting learning signals across domains (Fig. 2).

## 3.2 Dense Representation Learning

To serve as a general-purpose backbone, a time series encoder requires two distinct capabilities: detecting spatiotemporal structures and modelling temporal causality. To learn these from unlabelled data, we employ MDM, where a binary mask $\mathbf{m} \in \{0,1\}^{V_S \cdot T'}$ splits the input sequence $\mathbf{E}$ into visible tokens ($m_{v,t} = 1$) and masked tokens ($m_{v,t} = 0$). In standard MDM, the randomly masked tokens are reconstructed from the visible view, forcing the encoder to model spatiotemporal structures. However, random masking allows access to future context, which can impede the learning of temporal causality. We therefore propose *dual masking*, a strategy that alternates between two distinct distributions of $m_{v,t}$ during pre-training (Fig. 3).

**Structural Learning** In 75 % of training steps, we apply *random masking* where $m_{v,t} \sim \text{Bernoulli}(1 - \rho)$, with $\rho \in [0,1]$ denoting the masking ratio. This forces the encoder to learn the spatiotemporal structures necessary for accurate reconstruction.

**Causal Learning** In the remaining 25 % of steps, we apply *post-fix masking* where $m_{v,t} = \mathbb{1}_{[t \leq T'/2]}$. Hiding future context transforms the reconstruction task into a forecasting task, explicitly forcing the encoder to model temporal causality.

Once a mask $\mathbf{m}$ is drawn, it is applied to the input $\mathbf{E}$, yielding a reduced sequence $\mathbf{E}[\mathbf{m}] \in \mathbb{R}^{N_\text{v} \times D}$, where $N_\text{v}$ denotes the number of visible tokens. The encoder $f(\cdot)$ processes this sequence to produce latent time series features:

$$\mathbf{H} = f(\mathbf{E}[\mathbf{m}]) \in \mathbb{R}^{N_\text{v} \times D}. \quad (1)$$

To reconstruct the input, a full sequence $\mathbf{H}' \in \mathbb{R}^{(V_S \cdot T') \times D}$ is created by combining the latent features with learnable mask embeddings $\mathbf{e}^{\mathcal{M}} \in \mathbb{R}^{1 \times D}$ that are inserted at the masked positions. Crucially, we re-inject the positional embeddings to provide exact spatiotemporal context to the decoder:

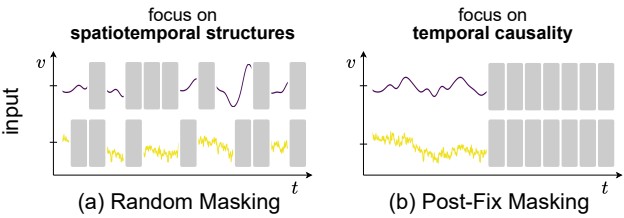

focus on **spatiotemporal structures**

focus on **temporal causality**

(a) Random Masking

(b) Post-Fix Masking

Figure 3: **Learning spatiotemporal structures and temporal causality** via (a) random masking and (b) post-fix masking.

$$\mathbf{h}'_{v,t} = \begin{cases} \mathbf{h}_{v,t} + \mathbf{e}_t^{\mathcal{T}} + \mathbf{e}_{S,v}^{\mathcal{V}} & \text{if } m_{v,t} = 1 \\ \mathbf{e}^{\mathcal{M}} + \mathbf{e}_t^{\mathcal{T}} + \mathbf{e}_{S,v}^{\mathcal{V}} & \text{if } m_{v,t} = 0 \end{cases} \quad \in \mathbb{R}^{1 \times D}. \quad (2)$$

The decoder $g(\cdot)$ then reconstructs the input patches:

$$\widehat{\mathbf{X}} = g(\mathbf{H}') \in \mathbb{R}^{V_S \times (T' \cdot P)}, \quad (3)$$

where $(T' \cdot P) = \overline{T}$ is the total context length in time points.

### 3.3 Structure-Aware Optimisation

Standard MDM typically optimises reconstruction using a mean squared error (MSE) loss, which measures the point-wise Euclidean distance between the input and the reconstruction:

$$\mathcal{L}_{\mathrm{MSE}} = \frac{1}{V_S \cdot T'} \sum_{v=1}^{V_S} \sum_{t=1}^{T'} \|\mathbf{x}_{v,t} - \widehat{\mathbf{x}}_{v,t}\|_2^2. \tag{4}$$

While effective for enforcing *local accuracy*, relying solely on MSE is suboptimal for learning robust time series features, as it treats time points independently and ignores global sequence structure. Consequently, reconstructions that capture the correct shape but suffer from amplitude scaling or offset shifts incur disproportionately high penalties. Furthermore, when faced with uncertainty in masked regions, MSE biases predictions toward the local mean (Mathieu et al., 2016; Isola et al., 2017; Rahaman et al., 2019), resulting in overly smooth reconstructions that lack structural fidelity.

To mitigate this, we introduce a *normalised cross-correlation* (NCC) loss. By measuring the linear dependence between the input and the reconstruction, it rewards relative alignment in phase and shape rather than strict absolute accuracy (Fig. 4):

$$\mathcal{L}_{\mathrm{NCC}} = \frac{1}{V_S \cdot \overline{T}} \sum_{v=1}^{V_S} \sum_{i=1}^{\overline{T}} \frac{(x_{v,i} - \mu_{\mathbf{x}_v})(\widehat{x}_{v,i} - \mu_{\widehat{\mathbf{x}}_v})}{\sigma_{\mathbf{x}_v} \sigma_{\widehat{\mathbf{x}}_v}}, \tag{5}$$

where $x_{v,i}$ denotes the value at the $i$-th time point of variate $v$, while $\mu$ and $\sigma$ represent the variate-level mean and standard deviation, respectively. Maximising NCC preserves the *global structure* and prevents potential mean-collapse. The final objective balances local accuracy and structural alignment:

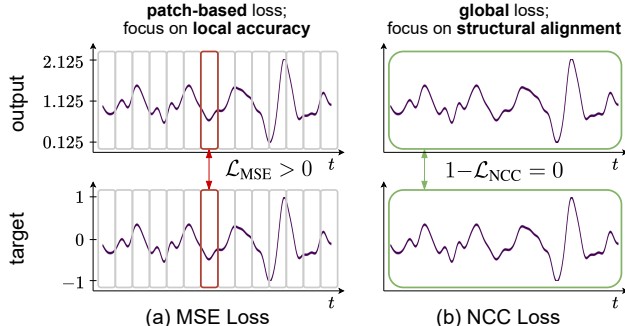

Figure 4: **Optimising local accuracy and global structure** using an (a) MSE loss and (b) NCC loss.

$$\mathcal{L} = \mathcal{L}_{\mathrm{MSE}} + \lambda \cdot (1 - \mathcal{L}_{\mathrm{NCC}}), \tag{6}$$

where $\lambda$ is a weighting hyperparameter. This composite objective allows the encoder to learn meaningful time series features without wasting modelling capacity on noise or scaling artifacts.

### 3.4 Versatile Downstream Deployment

The primary goal of `OTIS` is to extract high-quality time series features that serve as a robust foundation for diverse applications. For *discriminative tasks* like classification and regression, the pre-trained encoder $f(\cdot)$ serves as a powerful feature extractor. After processing the unmasked input $\mathbf{E}$, its output $\mathbf{H} = f(\mathbf{E}) \in \mathbb{R}^{(V_S \cdot T') \times D}$ is pooled via global averaging to obtain a dense representation $\mathbf{h}^* \in \mathbb{R}^{1 \times D}$, which is then passed to a simple linear layer for prediction. Optimisation is done using a cross-entropy loss for classification and an MSE loss for regression. Crucially, full-parameter fine-tuning of the encoder is computationally feasible due to its tiny 7.1 M size, enabling the extracted features $\mathbf{H}$ to be optimally adapted to the downstream application.

Additionally, these dense time series features can be seamlessly repurposed for simple *generative tasks*, such as short-term forecasting, at minimal additional computational cost. To this end, we stack the lightweight 1.5 M decoder $g(\cdot)$ retained from pre-training on top of the encoder $f(\cdot)$. For optimal downstream performance, they are fine-tuned end-to-end using post-fix masking and the composite loss defined in Eq. 6.

## 4 Experiments

**Architecture** OTIS is a tiny 7.1 M Transformer (Vaswani et al., 2017) encoder $f(\cdot)$ with 12 layers, a dimension of 192, an MLP size of 768, and 3 heads. For the masked data modelling objective, a lightweight 1.5 M Transformer decoder $g(\cdot)$ with 4 layers, a dimension of 160, an MLP size of 640, and 5 heads is employed. Patch size and stride are set to $P = 24$, splitting a time series into $T' = \overline{T}/P$ non-overlapping patches along the temporal axis. During pre-training, a context length of $\overline{T} = 1008$ time points is used, resulting in $T' = 42$ temporal embeddings. Crucially, for inference, the temporal embeddings are linearly interpolated to match any sequence length, providing the downstream flexibility to process time series of arbitrary duration.

**Data & Training** We curate a diverse pre-training corpus (640 k samples, 11 B time points; App. A) spanning healthcare, engineering, natural sciences, and finance. This diversity ensures OTIS is exposed to heterogeneous variate semantics and temporal dynamics. Particularly, to align with the growing interest in deploying general-purpose backbones for healthcare applications (Saab et al., 2024; Zhang et al., 2024), our corpus includes large datasets of the most widely used physiological signals, 12-lead ECG and 10-20 system EEG. We pre-train OTIS for 200 epochs on 4 NVIDIA A100 GPUs using AdamW (Loshchilov & Hutter, 2017) with batch size of 5120, masking ratio $\rho = 0.75$, NCC weight $\lambda = 0.1$ (App. F), and cosine annealing with learning rate of $3e$-5 and 10 % warm-up. Fine-tuning and inference run on a single NVIDIA A6000 GPU. Details on computational costs and hyperparameters are provided in Appendices B and C.

**Benchmarks** We evaluate OTIS on a total of 162 downstream tasks (App. D). To test its *discriminative capabilities*, we evaluate in two key areas. First, standard classification tasks including fault detection in rolling bearings (FD-B Lessmeier et al. (2016)), hand-gestures from accelerometers (Gesture Liu et al. (2009)), and muscular diseases from electromyography (EMG Goldberger et al. (2000)). We follow standard procedures from the original works and report accuracy (ACC ↑). Second, to specifically validate potential deployment on wearable health monitors, we extend the evaluation to real-world medical tasks. We further benchmark on 23 UEA (Bagnall et al., 2018) and 121 UCR (Dau et al., 2019) archive datasets, following standard protocols and reporting balanced accuracy (bACC ↑). Second, to specifically validate potential deployment on wearable health monitors, we extend the evaluation to real-world medical tasks. These include the detection of epileptic seizures (Epilepsy Andrzejak et al. (2001)), sleep stages (SleepEDF Goldberger et al. (2000)), and event types (TUEV Harati et al. (2015)) from EEG. We follow standard procedures from the original works and report balanced accuracy (bACC ↑). Further, we predict cardiac phenotypes such as the left ventricular (LV) end-diastolic volume (EDV), end-systolic volume (ESV), stroke volume (SV), ejection fraction (EF), and mass (M) from ECG (UK BioBank (Sudlow et al., 2015)). We follow standard protocols from Bai et al. (2024) and Turgut et al. (2025) and report R-squared ($R^2$ ↑). To explore OTIS' *generative capabilities*, we stack the lightweight 1.5 M decoder retained from pre-training on top of it and perform short-term forecasts (prediction and look-back window size of 96 and 336 time points, respectively). These include standard benchmarks on transformer temperature (ETT Zhou et al. (2021)), meteorology (Weather Wetterstation (2024)), household electricity consumption (Electricity UCI (2024)), influenza-like illness occurrence (ILI (CDC, 2024)), and road occupancy (Traffic PeMS (2024)). We follow established procedures from Zhou et al. (2021) and report normalised mean squared error (MSE ↓). Mean and standard deviation is reported across five seeds.

**Baselines** We compare OTIS against a comprehensive suite of 31 baselines (App. E). Primarily, we select state-of-the-art *general-purpose encoders* pre-trained on large heterogeneous corpora, specifically the 8.2 M UniTS (Gao et al., 2024) and 386 M MOMENT (Goswami et al., 2024). We also include *simple encoders* for general time series analysis like TS2Vec (Yue et al., 2022), TimesNet (Wu et al., 2022), TF-C (Zhang et al., 2022), Ti-MAE (Li et al., 2023), SimMTM (Dong et al., 2023), PatchTST (Nie et al., 2023), CroSSL (Deldari et al., 2024), TSLANet (Eldele et al., 2024), and InvConvNet (Germain et al., 2025). For the medical benchmarks, we compare against *specialised EEG encoders*, such as Handcrafted (Engemann et al., 2022), ShallowNet (Schirrmeister et al., 2017), DeepNet (Schirrmeister et al., 2017), EEGNet (Lawhern et al., 2018), CBraMod (Wang et al., 2025), and CSBrain (Zhou et al., 2025). Further, we evaluate against *specialised ECG encoders* trained on UK BioBank (Sudlow et al., 2015), the largest biomedical databank, including a supervised Transformer, ECG-Transformer, a Transformer pre-trained with masked data modelling (He et al., 2022), ECG-MAE, and Transformers trained multi-modally with paired imaging data, CM-AE (Radhakrishnan et al.,

Table 1: **Classification. Best** and second best scores (ACC ↑) are highlighted. `OTIS` demonstrates strong performance, outperforming specialised encoders as well as large general-purpose encoders.

(a) Standard benchmarks.

| Encoder | Machine fault | Hand gesture | Muscular disease |
|---|---|---|---|
| *Simple encoders* | | | |
| `TS2Vec` | $0.479 \pm 0.011$ | $0.692 \pm 0.033$ | $0.785 \pm 0.032$ |
| `TimesNet` | $0.619 \pm 0.020$ | $0.598 \pm 0.026$ | $0.912 \pm 0.021$ |
| `TF-C` | $0.694 \pm 0.023$ | $0.764 \pm 0.019$ | $0.817 \pm 0.029$ |
| `Ti-MAE` | $0.609 \pm 0.031$ | $0.719 \pm 0.023$ | $0.700 \pm 0.017$ |
| `SimMTM` | $0.685 \pm 0.042$ | $0.778 \pm 0.074$ | $0.951 \pm 0.000$ |
| `PatchTST` | $0.730 \pm 0.052$ | $0.617 \pm 0.000$ | $0.932 \pm 0.032$ |
| `TSLANet` | $0.756 \pm 0.020$ | $\mathbf{0.787} \pm 0.014$ | $0.951 \pm 0.000$ |
| `CroSSL` | $0.737 \pm 0.026$ | $0.610 \pm 0.014$ | $0.951 \pm 0.000$ |
| `InvConvNet` | $0.681 \pm 0.009$ | $0.458 \pm 0.031$ | $0.961 \pm 0.033$ |
| *General-purpose encoders* | | | |
| `MOMENT-386M` | $0.928$ $\pm 0.024$ | $0.618 \pm 0.017$ | $0.976$ $\pm 0.000$ |
| `UniTS-8.2M` | $0.749 \pm 0.052$ | $0.615 \pm 0.009$ | $0.971 \pm 0.011$ |
| `OTIS-7.1M` | $\mathbf{0.981} \pm 0.007$ | $0.780$ $\pm 0.020$ | $\mathbf{0.976} \pm 0.000$ |

(b) EEG benchmarks.

| Encoder | Epileptic seizure | Sleep stage | Event type |
|---|---|---|---|
| *Simple encoders* | | | |
| `PatchTST` | $0.861 \pm 0.021$ | $0.699 \pm 0.033$ | $0.531 \pm 0.212$ |
| `TSLANet` | $0.899 \pm 0.010$ | $0.726 \pm 0.011$ | $0.460 \pm 0.031$ |
| `CroSSL` | $0.862 \pm 0.022$ | $0.701 \pm 0.005$ | $0.380 \pm 0.036$ |
| `InvConvNet` | $0.854 \pm 0.040$ | $0.675 \pm 0.016$ | $0.449 \pm 0.024$ |
| *Specialised EEG encoders* | | | |
| `Handcrafted` | $\mathbf{0.930} \pm 0.000$ | $0.655 \pm 0.000$ | $0.311 \pm 0.000$ |
| `ShallowNet` | $0.901 \pm 0.021$ | $0.687 \pm 0.015$ | $0.493 \pm 0.026$ |
| `DeepNet` | $0.899 \pm 0.011$ | $0.740 \pm 0.007$ | $0.561 \pm 0.017$ |
| `EEGNet` | $0.872 \pm 0.019$ | $0.708 \pm 0.008$ | $0.474 \pm 0.015$ |
| `CBraMod` | $0.858 \pm 0.011$ | $0.735 \pm 0.004$ | $0.472 \pm 0.019$ |
| `CSBrain` | $0.855 \pm 0.029$ | $0.720 \pm 0.019$ | $0.522 \pm 0.030$ |
| *General-purpose encoders* | | | |
| `MOMENT-386M` | $0.918 \pm 0.007$ | $\mathbf{0.756} \pm 0.002$ | $0.517 \pm 0.044$ |
| `UniTS-8.2M` | $0.878 \pm 0.012$ | $0.710 \pm 0.011$ | $0.546$ $\pm 0.021$ |
| `OTIS-7.1M` | $0.918$ $\pm 0.014$ | $0.749$ $\pm 0.006$ | $\mathbf{0.614} \pm 0.013$ |

Table 2: **Regression. Best** and second best scores ($R^2$ ↑) are highlighted. `OTIS`' features are robust for critical healthcare applications such as cardiac phenotype regression from standard 12-lead electrocardiography.

| Encoder | LVEDV | LVESV | LVSV | LVEF | LVM |
|---|---|---|---|---|---|
| *Simple encoders* | | | | | |
| `PatchTST` | $0.321 \pm 0.006$ | $0.279 \pm 0.015$ | $0.234 \pm 0.006$ | $0.063 \pm 0.012$ | $0.369 \pm 0.042$ |
| `ECG-Transformer` | $0.394 \pm 0.008$ | $0.386 \pm 0.015$ | $0.294 \pm 0.011$ | $0.171 \pm 0.007$ | $0.459 \pm 0.021$ |
| `TSLANet` | $0.447 \pm 0.008$ | $0.421 \pm 0.007$ | $0.327 \pm 0.008$ | $0.173 \pm 0.008$ | $0.513 \pm 0.012$ |
| `CroSSL` | $0.451 \pm 0.004$ | $0.418 \pm 0.003$ | $0.315 \pm 0.003$ | $0.123 \pm 0.003$ | $0.506 \pm 0.002$ |
| `InvConvnet` | $0.369 \pm 0.001$ | $0.325 \pm 0.001$ | $0.267 \pm 0.001$ | $0.082 \pm 0.001$ | $0.430 \pm 0.002$ |
| *Specialised ECG encoders* | | | | | |
| `ECG-MAE` | $0.478 \pm 0.013$ | $0.475 \pm 0.010$ | $0.349 \pm 0.002$ | $0.233 \pm 0.016$ | $0.565 \pm 0.008$ |
| `CM-AE*` | $0.451 \pm 0.006$ | $0.380 \pm 0.019$ | $0.316 \pm 0.011$ | $0.103 \pm 0.012$ | $0.536 \pm 0.016$ |
| `MMCL*` | $0.498$ $\pm 0.012$ | $0.497$ $\pm 0.008$ | $0.360$ $\pm 0.013$ | $0.245$ $\pm 0.004$ | $0.597$ $\pm 0.018$ |
| *General-purpose encoders* | | | | | |
| `MOMENT-386M` | $0.449 \pm 0.012$ | $0.431 \pm 0.022$ | $0.344 \pm 0.014$ | $0.181 \pm 0.032$ | $0.532 \pm 0.012$ |
| `UniTS-8.2M` | $0.479 \pm 0.013$ | $0.458 \pm 0.011$ | $0.355 \pm 0.021$ | $0.197 \pm 0.024$ | $0.555 \pm 0.011$ |
| `OTIS-7.1M` | $\mathbf{0.506} \pm 0.002$ | $\mathbf{0.498} \pm 0.001$ | $\mathbf{0.386} \pm 0.004$ | $\mathbf{0.253} \pm 0.001$ | $\mathbf{0.604} \pm 0.002$ |

*Multi-modally pre-trained on paired time series and imaging data.

2023) and `MMCL` (Turgut et al., 2025). For the exploratory forecasting experiments, we include *simple models for forecasting* such as `N-BEATS` (Oreshkin et al., 2019) and `DLinear` (Zeng et al., 2023), as well as *specialised forecasting models* ranging from the 5 M `TTM` (Ekambaram et al., 2024) to the 2.4 B `Time-MoE` (Shi et al., 2025), alongside `GPT4TS` (Zhou et al., 2023), `MOIRAI` (Woo et al., 2024), `Chronos` (Ansari et al., 2024), `ROSE` (Wang et al., 2024), `Timer-XL` (Liu et al., 2025), and `DTAF` (Lu et al., 2026).

## 5 Results

### 5.1 `OTIS` Is Competitive with State of the Art *and* Highly Deployable

We first assess whether `OTIS` yields time series features that are (i) competitive with state-of-the-art specialised and general-purpose encoders and (ii) well-structured to serve any downstream task in any domain.

Table 3: **Prototypical 5-shot classification on the UEA and UCR archives. Best** and second best scores (bACC in % ↑) and wins (out of total datasets) are highlighted. Encoders are frozen after pre-training and class prototypes are formed from $k = 5$ random training samples per class; test samples are assigned by cosine similarity to the prototypes, averaged over 5 episodes with independently drawn training samples (App. G). OTIS attains the highest score on the UEA archive and remains competitive with the 54× larger MOMENT on the UCR archive, at a small fraction of its memory, energy, and latency cost.

(a) UEA archive (23 multi-variate datasets).

| Encoder | 5-Shot bACC | Wins | Memory [MB/Sample] ↓ | Energy [kWh/1M Samples] ↓ | Latency [ms/Sample] ↓ | Throughput [Samples/s] ↑ |
|---|---|---|---|---|---|---|
| MOMENT-386M | 51.5 ± 2.8 | 5 | 66.3 | 4.5 | 64.3 | 168.2 |
| UniTS-8.2M | 46.3 ± 3.0 | 2 | 4.9 | 0.1 | 0.8 | 5149.9 |
| OTIS-7.1M | **55.7** ± 2.6 | **16** | 11.5 | 0.1 | 1.4 | 2549.4 |

(b) UCR archive (121 uni-variate datasets).

| Encoder | 5-Shot bACC | Wins | Memory [MB/Sample] ↓ | Energy [kWh/1M Samples] ↓ | Latency [ms/Sample] ↓ | Throughput [Samples/s] ↑ |
|---|---|---|---|---|---|---|
| MOMENT-386M | **60.9** ± 3.3 | **63** | 27.3 | 0.437 | 6.3 | 492.2 |
| UniTS-8.2M | 52.2 ± 3.5 | 11 | 0.9 | 0.005 | 0.1 | 12238.2 |
| OTIS-7.1M | 59.8 ± 3.4 | 47 | 1.1 | 0.014 | 0.3 | 3850.6 |

**Discriminative Capabilities** On standard benchmarks (Tab. 1a), OTIS shows strong discriminative capabilities, consistently outperforming the general-purpose encoders UniTS and MOMENT. Notably, it achieves state-of-the-art on FD-B despite the dataset containing long time series ($T = 5120$). As OTIS was pre-trained with $\overline{T} = 1008$, this confirms that our linear interpolation of temporal embeddings generalises to unseen resolutions and sequence lengths. The encoder adapts seamlessly to novel domains such as inertial measurement units in wearables (Gesture) and single-channel EMG, suggesting that it does not overfit to the distributions of the pre-training data but rather learns a meaningful representation space. This quality extends to medical benchmarks, particularly electroencephalography (EEG), where OTIS outperforms specialised EEG encoders (Tab. 1b). Further, on electrocardiography (ECG) regression (Tab. 2), it consistently surpasses specialised ECG encoders such as MMCL and CM-AE which use multi-modal supervision. These results confirm that OTIS captures predictive *spatiotemporal structures* robust enough for critical healthcare applications.

**Few-Shot Capabilities** To evaluate the quality of the time series features learned during pre-training, we perform prototypical k-shot classification with *frozen encoders* and *no task-specific fine-tuning* (App. G) on a total of 144 unseen datasets: 23 from the UEA multi-variate archive (Bagnall et al., 2018) and 121 from the UCR uni-variate archive (Dau et al., 2019). To this end, $k = 5$ labelled training samples are randomly sampled per class, their embeddings are averaged into a per-class prototype, and then test samples are classified by nearest prototype using cosine similarity. To support robustness of the findings, this procedure is repeated for 5 episodes, each with independently drawn training samples. On the UEA archive (Tab. 3a), OTIS achieves the best performance (55.7 % vs 51.5 % MOMENT, 46.3 % UniTS), winning on 16 of 23 datasets, while consuming 6× less memory (11.5 vs 66.3 MB/sample), 45× less energy (0.1 vs 4.5 kWh/1M samples), and operating at 46× lower latency (1.4 vs 64.3 ms/sample) than MOMENT. On the UCR archive (Tab. 3b), it remains competitive with MOMENT (59.8 % vs 60.9 %) and clearly outperforms UniTS (52.2 %), at 25× lower memory (1.1 vs 27.3 MB/sample), 31× less energy (0.014 vs 0.437 kWh/1M samples), and 20× lower latency (0.3 vs 6.3 ms/sample). Notably, OTIS is the only encoder that excels in *both* regimes: it dominates full fine-tuning while remaining competitive under severe label scarcity, confirming that its representations are both expressive and well-structured to generalise on any downstream task in any domain.

**Generative Capabilities** We investigate if the utility of OTIS' time series representations extends to simple generative tasks. By attaching the lightweight 1.5 M decoder obtained from pre-training, we evaluate on standard short-term forecasting (Tab. 4). OTIS performs competitively, outperforming UniTS and MOMENT on 5 of 8 datasets. Surprisingly, despite not being a specialised forecasting model, it achieves the lowest errors

Table 4: **Forecasting. Best** and second best scores (MSE ↓) are highlighted. The utility of `OTIS`' time series features seamlessly extends to simple generative tasks such as short-term forecasting.

| Encoder | ETTh1 | ETTh2 | ETTm1 | ETTm2 | Weather | Electricity | Illness | Traffic |
|---|---|---|---|---|---|---|---|---|
| *Simple forecasting models* | | | | | | | | |
| N-BEATS | 0.399 | 0.327 | 0.318 | 0.197 | 0.152 | 0.131 | 4.539 | 0.375 |
| DLinear | 0.375 | 0.289 | 0.299 | 0.167 | 0.176 | 0.140 | 2.215 | 0.410 |
| PatchTST | 0.370 | 0.274 | 0.293 | 0.166 | 0.149 | 0.129 | **1.319** | 0.360 |
| *Specialised forecasting models* | | | | | | | | |
| TTM | 0.363 | 0.262 | 0.283 | **0.158** | 0.149 | 0.128 | 2.682 | 0.352 |
| GPT4TS | 0.376 | 0.285 | 0.292 | 0.173 | 0.162 | 0.139 | 2.063 | 0.388 |
| ROSE | 0.362 | 0.283 | 0.291 | 0.164 | 0.159 | 0.138 | 2.814 | 0.377 |
| MOIRAI | 0.375 | 0.277 | 0.335 | 0.189 | 0.167 | 0.152 | 2.923 | 0.395 |
| Chronos | 0.427 | 0.303 | 0.391 | 0.194 | 0.183 | 0.165 | 3.142 | 0.459 |
| Timer-XL | 0.369 | 0.278 | 0.290 | 0.175 | 0.157 | 0.127 | 2.835 | **0.340** |
| Time-MoE | **0.344** | 0.275 | **0.271** | 0.179 | 0.159 | 0.147 | 2.673 | 0.367 |
| DTAF | 0.382 | 0.289 | 0.306 | 0.175 | 0.149 | 0.143 | 2.163 | 0.386 |
| *General-purpose encoders* | | | | | | | | |
| MOMENT-386M | 0.387 | 0.288 | 0.293 | 0.170 | 0.154 | 0.136 | 2.728 | 0.391 |
| UniTS-8.2M | 0.394 | 0.361 | 0.369 | 0.259 | 0.216 | 0.152 | 2.472 | 0.421 |
| OTIS-7.1M | 0.419 | **0.247** | 0.337 | 0.162 | **0.139** | **0.125** | 2.643 | 0.387 |

on 3 of 8 datasets, surpassing even the 2.4 B forecasting foundation model `Time-MoE`. This indicates that `OTIS` encodes *temporal dynamics* rich enough for extrapolating short future horizons from past observations.

To further validate this, we perform a controlled experiment on sine waves with both `OTIS` and the 1.5 M decoder completely frozen after pre-training (Fig. 5). We feed `OTIS` a 50 Hz sine wave covering time steps $t \in [0, T_{\text{input}}]$ and pass the extracted features to the decoder to predict the remaining sequence at steps $t \in [T_{\text{input}} + 1, T_{\text{forecast}}]$. The decoder faithfully reproduces the original input frequency (Fig. 5a), confirming that the frozen time series features capture relevant temporal information. While long-horizon generations can suffer from gradual mean-collapse, we show this can be mitigated by fine-tuning *solely* the learnable variate embedding within the tokeniser (Fig. 5b). This noticeably improves the expressiveness of `OTIS`' frozen time series features, even enabling the decoder to reproduce frequencies of unseen sine waves (Fig. 5c). These experiments confirm that `OTIS` learns well-structured representations that generalise beyond the distribution of the training data.

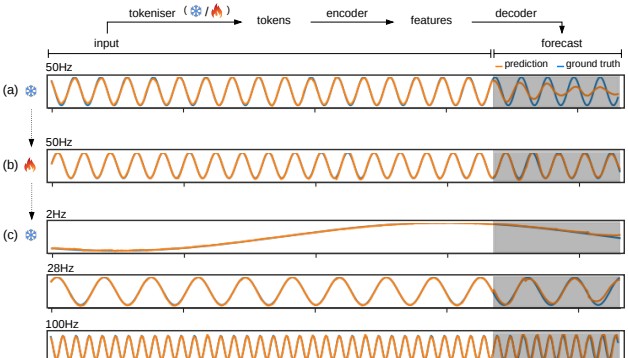

Figure 5: **Frozen time series representations encode temporal patterns.** `OTIS` and the 1.5 M decoder from pre-training are *frozen*. (a) The decoder reproduces the input frequency, confirming that `OTIS` embeds relevant temporal information in its features. (b) Fine-tuning *solely* the variate embedding to learn the correct semantics of a 50 Hz sine wave counteracts the mean-collapse emerging over longer horizons, without updating any encoder weights. (c) This even enables the decoder to correctly reproduce frequencies of unseen sine waves, confirming that `OTIS`' representations generalise beyond the training distribution.

## 5.2 `OTIS`
### Learns a Well-Structured Representation Space

To understand *why* `OTIS` generalises across tasks and domains, we inspect its learned representation space.

**Addressing Multi-Domain Data Heterogeneity** Different domains have unique semantics especially across variates. If these semantic differences are not addressed, they become conflicting learning signals for the encoder. We find that learnable variate embeddings successfully address this heterogeneity (Fig. 6).

*Neurology.* EEG electrodes are spatially arranged according to the standard 10-20 system (Homan et al., 1987). The learned variate embeddings successfully recover this underlying spatial topology (Fig. 6a). We

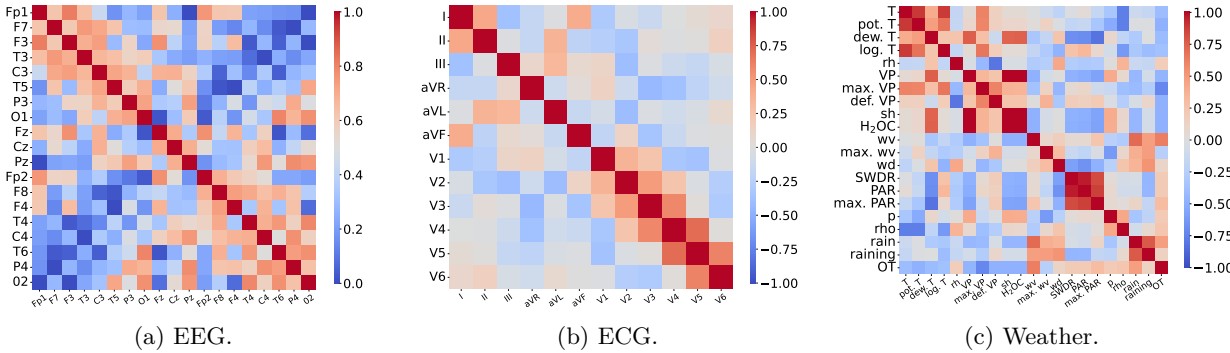

Figure 6: **Addressing multi-domain data heterogeneity.** Cosine similarities of variate embeddings $\mathbf{e}^{\mathcal{V}}_{S,v}$ show variate semantics in (a) EEG, (b) ECG, and (c) Weather are correctly modelled by the tokeniser.

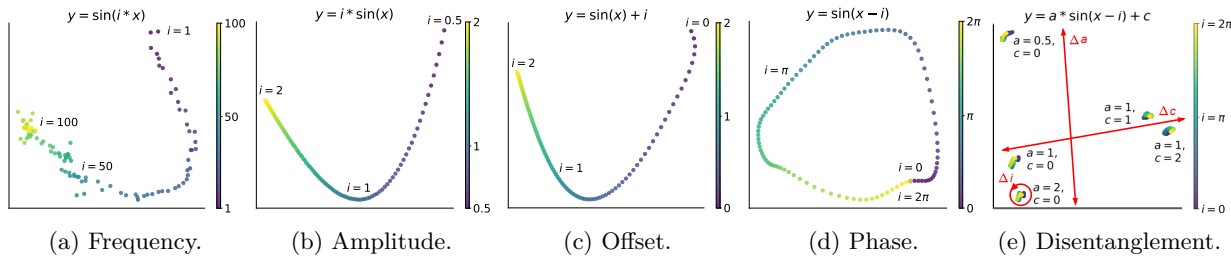

Figure 7: **Learning a consistent representation space.** Principal component analysis of *zero-shot* sine wave representations. (a-c) Continuous properties map to logarithmic manifolds. (d) Phase maps to a closed circle. (e) Simultaneous changes in these properties are orthogonally disentangled in the representation space.

observe strong similarity among electrodes within the left hemisphere (Fp1 to O1), right hemisphere (Fp2 to O2), and the sagittal plane (Fz, Cz, Pz). Conversely, low similarity accurately corresponds to electrodes with anatomically distant separation.

*Cardiology.* Similarly, the learned variate embeddings for 12-lead ECG reflect the standard acquisition protocol (Fig. 6b). We find high similarity among the precordial leads V1–V6 (Wilson et al., 1934), which physically span the rib cage. Furthermore, the limb leads I, II, and III (Einthoven et al., 1902) exhibit similarities consistent with the Einthoven triangle (Kligfield et al., 2007).

*Meteorology.* The learned variate embeddings for climatological indicators cluster by thermodynamic category: temperature (T), humidity (rh, VP, sh, $H_2OC$), wind (wv, wd), radiation (SWDR, PAR), pressure (p, rho), and precipitation (rain, raining) (Fig. 6c). Correlations across categories are also captured, such as the thermodynamic law def. $VP \propto (1 - rh)$ (Shamshiri et al., 2018) between vapor pressure deficit (def. VP in mbar) and relative humidity (rh in %), and the more intuitive inverse relationship between precipitation (raining in seconds) and solar radiation (SWDR in $W/m^2$).

**Learning a Consistent Representation Space**    With multi-domain data heterogeneity addressed by the tokeniser, the encoder is free to learn a well-structured, consistent representation space. To validate this, we freeze `OTIS` after pre-training and run controlled experiments on sine waves with distinct properties (Fig. 7).

*Manifold Geometry.* To explore the structure of the learned representation space, we probe `OTIS` using sine waves with varying frequency, amplitude, phase, and offset. We find that the representation space exhibits a well-defined geometry for these time series properties. For frequency (Fig. 7a), the representations follow a logarithmic mapping, where high frequencies (e.g. 50–100 Hz) are clustered more tightly than low frequencies (e.g. 1–50 Hz). This aligns with intuition: an absolute increase of 1 Hz from 5 to 6 Hz reflects a relative change of 20 %, whereas an increase from 50 to 51 Hz reflects only 2 %. Similarly, amplitude (Fig. 7b) and offset (Fig. 7c) follow logarithmic mappings. Most notably, the mapping of phase (Fig. 7d) forms a closed circle.

Table 5: **Inductive bias is essential.** A leave-one-out ablation evaluates the average performance gain (PG) contributed by the novel components. The components are mutually reinforcing, with the removal of any single bias leading to suboptimal performance. Structure-aware optimisation using a normalised cross-correlation loss supplies the largest individual gains.

| Encoder | Classification | | Regression | | Forecasting | |
|---|---|---|---|---|---|---|
| | ACC | PG | $R^2$ | PG | MSE | PG |
| `OTIS` | 85.7 | | 0.442 | | 0.976 | |
| w/o domain-aware tokenisation | 84.5 | ↑ 1.42 % | 0.441 | ↑ 0.23 % | 1.075 | ↑ 9.21 % |
| w/o dual masking | 84.1 | ↑ 1.90 % | 0.443 | ↓ 0.23 % | 0.995 | ↑ 1.91 % |
| w/o normalised cross-correlation loss | 83.5 | ↑ 2.63 % | 0.438 | ↑ 0.91 % | 1.161 | ↑ 15.93 % |
| w/o pre-training | 75.1 | ↑ 14.11 % | 0.322 | ↑ 37.27 % | 2.077 | ↑ 53.00 % |

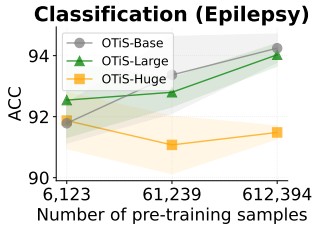
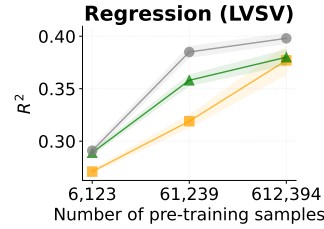
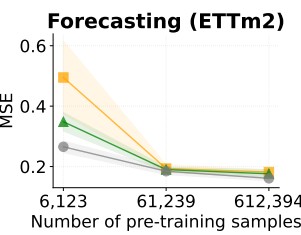

Figure 8: **Scaling is not trivial.** Downstream performance scales reliably with the size of the pre-training corpus (1 %, 10 %, 100 %), but saturates with encoder size: Large (40 M) and Huge (116 M) variants offer little advantage over the 7.1 M base. Shaded regions indicate standard deviation across five seeds.

The embedding for $\phi = 0$ is identical to $\phi = 2\pi$ and opposed to $\phi = \pi$, where $\phi \in [0, 2\pi]$ denotes the phase. This topological closure correctly mirrors the unit circle $e^{i\phi} = \cos(\phi) + i\sin(\phi)$. Consequently, this continuous and topologically consistent representation space proves that the encoder learns the *actual* structure of these properties rather than memorising distributions of the pre-training data. This empowers `OTIS` to generalise to unseen time series, making it a robust general-purpose encoder despite its tiny size.

*Orthogonal Disentanglement.* Crucially, when multiple properties vary simultaneously, the encoder disentangles them into orthogonal subspaces of the representation space (Fig. 7e). Changes in amplitude do not distort the representation with respect to phase or offset, suggesting these properties are treated as independent variables. This orthogonality simplifies downstream adaptation of `OTIS`, allowing task-specific heads to isolate relevant signal properties (e.g. amplitude) without interference from irrelevant variations (e.g. phase shifts).

### 5.3 `OTIS` Benefits from Targeted Inductive Bias

We conduct a comprehensive ablation study to quantify the impact of our three proposed inductive biases: (1) the domain-aware tokeniser, (2) the dual masking strategy, and (3) the structure-aware objective. First, replacing the domain-aware tokeniser with a domain-agnostic one (i.e. universal positional embeddings shared across all domains) consistently degrades performance across all tasks. This confirms that explicitly resolving semantic heterogeneity is a prerequisite for effective representation learning in multi-domain settings. Second, reverting from dual to purely random masking harms performance, particularly in forecasting. This validates our hypothesis that forcing the encoder to predict future horizons during pre-training is critical for learning temporal patterns. Third, removing the NCC loss term from the training objective causes the most significant performance decline. This underscores that relying solely on local point-wise reconstruction via MSE is insufficient; explicitly enforcing global structural alignment is essential for learning robust time series representations. Overall, these results demonstrate that the proposed components are mutually reinforcing, and that the interplay of all three is essential to maximise the performance of a tiny encoder.

### 5.4 `OTIS` **Scales with Data**

Having shown that our simple pre-training recipe enables standard masked data modelling to produce robust features, we now examine its scalability. To this end, we analyse the scaling behaviour of `OTIS` with respect to both dataset and encoder size (Fig. 8). We train variants using subsamples of the pre-training data (1 %, 10 %, 100 %) and introduce two larger configurations of our encoder: Large (40 M) and Huge (116 M) (App. C). Our analysis reveals two distinct trends. First, *performance scales reliably with data*. We observe consistent downstream gains as the pre-training corpus grows, indicating that the encoder effectively leverages larger datasets to refine its representation space. Second, *performance saturates with encoder size*. We find that the Large and Huge encoder variants offer almost no performance advantage over the tiny 7.1 M variant. This plateau suggests that for time series, which often exhibit large redundancy especially in multi-variate settings, producing arbitrarily large, powerful encoders is non-trivial and requires sophisticated approaches. In natural language processing and computer vision, naive scaling is often hindered by training instabilities and feature collapse, requiring extensive engineering to mitigate (Siméoni et al., 2025; TeamOlmo, 2025). This is also in line with recent studies on time series analysis, which show that scaling is not as simple as increasing parameter counts (Woo et al., 2024; Goswami et al., 2024; Liu et al., 2026).

## 6 Discussion & Conclusion

In this work, we introduce `OTIS` to challenge the prevailing assumption that powerful general-purpose encoder require massive scaling. By strategically injecting inductive biases through (1) a *domain-aware tokeniser*, (2) a *dual masking strategy*, and (3) a *structure-aware objective*, we show that a tiny 7.1 M encoder can match the capabilities of state-of-the-art encoders orders of magnitude larger, while saving up to 98 % memory and 92 % training time. Our extensive evaluation yields three critical insights. First, `OTIS` learns a topologically consistent representation space that enables robust generalisation across tasks and domains, from industrial fault detection to clinical diagnostics. Second, the proposed inductive biases are mutually reinforcing; removing any single component notably degrades performance, confirming their collective necessity for learning high-quality time series features. Third, we observe that while performance scales reliably with data size, simple parameter scaling of time series encoders yields diminishing returns. Overall, these findings suggest that targeted inductive bias offers a promising alternative to the development of general-purpose encoders orthogonal to traditional scaling. By enabling robust deployment on resource-constrained devices, ranging from wearable health monitors to industrial edge sensors, `OTIS` marks a distinct step towards democratising access to high-quality time series features for any downstream application. To foster further research and deployment, we release our entire code base and pre-trained weights.

### Limitations

While `OTIS` matches state-of-the-art performance with a fraction of the parameters, our study surfaces several open challenges. First, performance scales reliably with data but saturates with encoder size: Large (40 M) and Huge (116 M) variants offer little advantage over the 7.1 M base, indicating that scaling time series encoders requires methodological advances beyond raw parameter counts. Second, unlike in NLP and CV, general-purpose time series encoders still rely on manually curated data; automated pipelines to filter and categorise time series from the web are a natural next step. Third, our evaluation is restricted to numeric time series, leaving the integration of other modalities, such as imaging or text, to future work.

### Broader Impact Statement

This work aims to advance the field of Time Series Analysis. By developing tiny general-purpose time series encoders that can in future be deployed on resource-constrained devices like wearables and industrial sensors, our work has potential positive societal impacts in democratising access to high-quality time series features. However, as with any general-purpose backbone, there are risks associated with deployment in safety-critical domains without sufficient validation. We believe further research into the robustness and interpretability of such tiny encoders is necessary before deployment, particularly in clinical and industrial routine.

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

# A    Pre-Training Corpus Details

In this section, we present an overview of our large and diverse pre-training corpus, as summarised in Table 6. The corpus consists of publicly available data spanning eight domains, with a total of $640, 187$ samples and 11 billion time points. In the following, we provide a detailed breakdown of the domains and the datasets they encompass. Note that we apply variate-wise standard normalisation to the datasets unless otherwise specified.

**ECG**    The MIMIC-IV-ECG dataset (Gow et al., 2023) contains diagnostic 10-second, 12-lead ECG recordings sampled at a frequency of $500\,Hz$. While the entire dataset comprises $800, 035$ samples, we include only the first half of the recordings available in the database, preventing the ECG data from predominating in the pre-training corpus. To remove the baseline drift from the ECG data, we use the asymmetric least square smoothing technique (Zhang et al., 2010). Note that we apply standard normalisation separately to the Einthoven, Goldberger, and Wilson leads.

**Temperature**    The Deutscher Wetterdienst (DWD) dataset (Wetterdienst, 2024) contains hourly air temperature measurements from 629 weather stations across Germany. Since the recording length varies significantly, ranging from 763 to $1, 148, 290$ hours per station, we split the data into chunks of 720 hours (approximately one month).

**Audio**    The AudioSet dataset (Gemmeke et al., 2017) contains 10-second YouTube clips for audio classification, featuring 527 types of audio events that are weakly annotated for each clip. The full training set includes a class-wise balanced subset (AudioSet-20K, $22, 176$ clips) and an unbalanced (AudioSet-2M $2, 042, 985$ clips) set. For our pre-training corpus, we use the balanced AudioSet-20K, which contains $3, 491$ mono and $16, 123$ stereo recordings, all sampled at $44.1\,kHz$.

**Electromechanics**    The FD-A dataset (Lessmeier et al., 2016) collects vibration signals from rolling bearings in a mechanical system for fault detection purposes. Each sample consists of $5, 120$ timestamps, indicating one of three mechanical device states. The FD-B dataset is similar to FD-A but includes rolling bearings tested under different working conditions, such as a distinct rotational speed, load torque, and radial force.

**EEG**    The TDBrain dataset (Van Dijk et al., 2022) includes raw resting-state EEG data from $1, 274$ psychiatric patients aged 5 to 89, collected between 2001 and 2021. The dataset covers a range of conditions, including Major Depressive Disorder (426 patients), Attention Deficit Hyperactivity Disorder (271 patients), Subjective Memory Complaints (119 patients), and Obsessive-Compulsive Disorder (75 patients). The data was recorded at $500\,Hz$ using 26 electrode EEG.

The SEED dataset (Zheng & Lu, 2015) contains EEG data recorded under three emotional states: positive, neutral, and negative. It comprises EEG data from 15 subjects, with each subject participating in experiments twice, several days apart. The data is sampled at $200\,Hz$ and recorded using 62 electrode EEG.

For simplicity, we only consider the 19 electrodes common to both datasets, i.e. the electrodes that correspond to the 10-20 electrode international system.

**Banking**    The NN5 competition dataset (Taieb et al., 2012) consists of daily cash withdrawals observed at 111 randomly selected automated teller machines across various locations in England.

**Economics**    The FRED-MD dataset (McCracken & Ng, 2016) contains 107 monthly time series showing a set of macro-economic indicators from the Federal Reserve Bank of St Louis. The data was extracted from the FRED-MD database.

The Exchange dataset (Lai et al., 2018) records the daily exchange rates of eight different nations, including Australia, Great Britain, Canada, Switzerland, China, Japan, New Zealand, and Singapore, ranging from 1990 to 2016.

Table 6: The pre-training data covers domains in healthcare, engineering, natural sciences, and finance.

| Domain | Name | Samples | Variates | Time points | Frequency |
|---|---|---|---|---|---|
| ECG | MIMIC-IV-ECG | $400,000$ | 12 | $5,000$ | $500\,\mathrm{Hz}$ |
| Temperature | DWD | $203,340$ | 1 | 720 | $278\,\mu\mathrm{Hz}$ |
| Audio (stereo) | AudioSet-20K | $16,123$ | 2 | $441,000$ | $44.1\,\mathrm{kHz}$ |
| Audio (mono) | AudioSet-20K | $3,491$ | 1 | $441,000$ | $44.1\,\mathrm{kHz}$ |
| Electromechanics | FD-A | $13,640$ | 1 | $5,120$ | $64\,\mathrm{kHz}$ |
| EEG | TDBrain | $2,692$ | 19 | $60,000$ | $500\,\mathrm{Hz}$ |
| | SEED | 675 | 19 | $37,000$ | $200\,\mathrm{Hz}$ |
| Banking | NN5 | 111 | 1 | 971 | $12\,\mu\mathrm{Hz}$ |
| Economics | FRED-MD | 107 | 1 | 728 | $386\,\mathrm{nHz}$ |
| | Exchange | 8 | 1 | $7,588$ | $12\,\mu\mathrm{Hz}$ |
| | | **640,187** | | **11,052,756,981** | |

# B   Computational Details

We provide an overview of the computational resources used for pre-training in Table 7. Details on pre-training the larger encoder configurations, Large and Huge, described in Section C.1, are also included.

Table 7: Computational resources used for pre-training.

| Method | Parameters | Power consumption | CPU count | GPU | | |
|---|---|---|---|---|---|---|
| | | | | Count | Hours | Type |
| OTIS | $7.1\,\mathrm{M}$ | $700\,\mathrm{W}^*$ | 128 | 4 | $86^\dagger$ | NVIDIA A100-80GB |
| OTIS-Large | $40\,\mathrm{M}$ | $800\,\mathrm{W}^*$ | 128 | 4 | $116^\dagger$ | NVIDIA A100-80GB |
| OTIS-Huge | $116\,\mathrm{M}$ | $960\,\mathrm{W}^*$ | 128 | 4 | $164^\dagger$ | NVIDIA A100-80GB |

\* Total power consumption across all GPUs.

$\dagger$ Total hours across all GPUs.

# C   Experimental Details

We outline encoder configurations in Section C.1 and hyperparameter tuning in Section C.2.

## C.1   Encoder Configurations

To explore the scaling laws with respect to the encoder size in Section 5.4, we introduce two larger configurations of our encoder $f(\cdot)$, Large and Huge, as summarised in Table 8.

Table 8: Details of encoder configurations.

| Method | Layers | Hidden size $D$ | MLP size | Heads | $d_{kv}$ | Parameters |
|---|---|---|---|---|---|---|
| OTIS | 12 | 192 | 768 | 3 | 64 | $7.1\,\mathrm{M}$ |
| OTIS-Large | 18 | 384 | 1536 | 6 | 64 | $40\,\mathrm{M}$ |
| OTIS-Huge | 24 | 576 | 2304 | 8 | 72 | $116\,\mathrm{M}$ |

## C.2   Pre-Training & Fine-Tuning Parameters

We provide the hyperparameters used for pre-training in Table 9. Fine-tuning hyperparameters for classification, regression, and forecasting tasks are summarised in Table 10, 11, and 12, respectively. These

Table 9: Hyperparameters for pre-training. Pre-training is run on 4 NVIDIA A100-80GB GPUs using a cosine learning rate scheduler with a 10% warm-up. All configurations employ a lightweight 1.5 M decoder consisting of 4 layers, a hidden size of 160, an MLP size of 640, and 5 heads.

| Method | Epochs | Batch size | Base LR | NCC $\lambda$ | Mask ratio $\rho$ | Weight decay |
|---|---|---|---|---|---|---|
| OTIS | 200 | 5120 | 3e-5 | 0.1 | 0.75 | 0.10 |
| OTIS-Large | 200 | 3328 | 1e-5 | 0.1 | 0.75 | 0.15 |
| OTIS-Huge | 200 | 2880 | 3e-6 | 0.1 | 0.75 | 0.05 |

Table 10: Hyperparameters used for fine-tuning the classification tasks on a single NVIDIA RTX A6000-48GB GPU. A cosine learning rate scheduler with a 10% warm-up is applied.

| Dataset | Method | Epochs | Batch size | Base LR | Drop path | Layer decay | Weight decay | Label smoothing |
|---|---|---|---|---|---|---|---|---|
| FD-B | OTIS | 75 | 32 | 3e-4 | 0.0 | 0.75 | 0.1 | 0.1 |
| Gesture | OTIS | 75 | 32 | 3e-3 | 0.2 | 0.50 | 0.1 | 0.1 |
| EMG | OTIS | 75 | 32 | 1e-3 | 0.2 | 0.75 | 0.1 | 0.2 |
| Epilepsy | OTIS | 75 | 32 | 3e-4 | 0.2 | 0.75 | 0.0 | 0.0 |
| SleepEDF | OTIS | 20 | 4352 | 1e-4 | 0.1 | 0.75 | 0.2 | 0.2 |
| TUEV | OTIS | 20 | 1216 | 3e-5 | 0.2 | 0.75 | 0.1 | 0.1 |

Table 11: Hyperparameters used for fine-tuning the regression tasks on a single NVIDIA RTX A6000-48GB GPU. A cosine learning rate scheduler with a 10% warm-up is applied.

| Dataset | Method | Epochs | Batch size | Base LR | Drop path | Layer decay | Weight decay |
|---|---|---|---|---|---|---|---|
| UK BioBank | OTIS | 50 | 192 | 3e-4 | 0.2 | 0.75 | 0.1 |

Table 12: Hyperparameters used for fine-tuning the forecasting tasks on a single NVIDIA RTX A6000-48GB GPU. A cosine learning rate scheduler with a 10% warm-up is applied.

| Dataset | Method | Epochs | Batch size | Base LR | NCC $\lambda$ | Weight decay |
|---|---|---|---|---|---|---|
| ETTh1 | OTIS | 1000 | 1 | 3e-2 | 0.1 | 0.15 |
| ETTh2 | OTIS | 1000 | 1 | 3e-2 | 0.2 | 0.25 |
| ETTm1 | OTIS | 1000 | 1 | 1e-2 | 0.2 | 0.25 |
| ETTm2 | OTIS | 1000 | 1 | 1e-2 | 0.1 | 0.25 |
| Weather | OTIS | 1000 | 1 | 1e-2 | 0.2 | 0.25 |
| Electricity | OTIS | 250 | 32 | 1e-3 | 0.0 | 0.25 |
| Illness | OTIS | 1000 | 1 | 3e-1 | 0.1 | 0.25 |
| Traffic | OTIS | 100 | 64 | 1e-2 | 0.0 | 0.05 |

optimal hyperparameters are found through a grid search over the learning rate (1e−6, 3e−6, ..., 1e−1, 3e−1), batch size ($2^x$, $x \in [2, 3, ..., 7]$), drop path (0.0, 0.1, 0.2), layer decay (0.5, 0.75), weight decay (0.0, 0.05, 0.1, 0.15, 0.2, 0.25), label smoothing (0.0, 0.1, 0.2) for classification, and NCC loss weight $\lambda$ (0.0, 0.1, 0.2) for forecasting.

Table 13: Summary of all datasets used for the evaluation, including details on the metrics, domains, and time series properties.

| Task | Metric | Dataset | | | | | |
|---|---|---|---|---|---|---|---|
| | | Domain $S$ | Name | Samples | Variates $V_S$ | Time points | Frequency |
| Classification | ACC | Electromechanics | FD-B 2016 | $13,640$ | 1 | $5,120$ | $64\,\text{kHz}$ |
| | | Acceleration | Gesture 2009 | 560 | 3 | 206 | $100\,\text{Hz}$ |
| | | EMG | EMG 2000 | 204 | 1 | $1,500$ | $4\,\text{kHz}$ |
| | | | Epilepsy 2001 | $11,500$ | 1 | 178 | $174\,\text{Hz}$ |
| | | EEG | SleepEDF 2000 | $195,479$ | 1 | $3,000$ | $100\,\text{Hz}$ |
| | | | TUEV 2016 | $112,237$ | 19 | $1,000$ | $200\,\text{Hz}$ |
| Regression | $R^2$ | ECG | UK BioBank 2015 | $18,926$ | 12 | $5,000$ | $500\,\text{Hz}$ |
| Forecasting | MSE | Energy | ETTh1 2021 | 1 | 7 | $17,420$ | (hourly) $278\,\mu\text{Hz}$ |
| | | | ETTh2 2021 | 1 | 7 | $17,420$ | (hourly) $278\,\mu\text{Hz}$ |
| | | | ETTm1 2021 | 1 | 7 | $69,680$ | (minutely) $1.1\,\text{mHz}$ |
| | | | ETTm2 2021 | 1 | 7 | $69,680$ | (minutely) $1.1\,\text{mHz}$ |
| | | Weather | Weather 2024 | 1 | 21 | $52,696$ | (minutely) $2.8\,\text{mHz}$ |
| | | Electricity | Electricity 2024 | 321 | 1 | $26,304$ | (hourly) $278\,\mu\text{Hz}$ |
| | | Disease | Illness 2024 | 1 | 7 | 966 | (weekly) $1.6\,\mu\text{Hz}$ |
| | | Logistics | Traffic 2024 | 862 | 1 | $17,544$ | (hourly) $278\,\mu\text{Hz}$ |

## D  Benchmark Details

Table 13 provides an overview of all benchmarks conducted in this study.

## E  Baseline Details

Table 14 provides an overview of all baseline methods used in this study.

## F  Additional Results: Effect of NCC Loss on Time Series Feature Quality

We ablate the NCC loss weight $\lambda$ by evaluating 5-shot prototypical classification across 6 benchmark datasets. Table 15 reports balanced accuracy for $\lambda \in \{0.0, 0.1, 0.2\}$. Setting $\lambda = 0.1$ improves the average by $+2.0\,\text{pp}$ over $\lambda = 0$ ($67.0\,\%$ vs $65.0\,\%$), with gains on 5 of 6 datasets while most notably Epilepsy ($+5.5\,\text{pp}$) and hand gesture ($+2.8\,\text{pp}$); only sleep stage drops marginally ($-0.8\,\text{pp}$). Increasing to $\lambda = 0.2$ further benefits hand gesture ($+5.8\,\text{pp}$ over $\lambda = 0$) and Epilepsy ($+7.1\,\text{pp}$) but degrades sleep stage ($-2.6\,\text{pp}$) and muscular disease ($-1.1\,\text{pp}$). These trends are consistent with the full-training ablation (Tab. 5), where NCC removal causes the largest performance drop, and motivate the choice of $\lambda = 0.1$ used throughout the study.

## G  Additional Results: Prototypical Few-Shot Classification

**Evaluation protocol.**  We evaluate each encoder (`OTIS`, `MOMENT`, `UniTS`) under a *prototypical $k$-shot* classification protocol, directly after pre-training and without any task-specific fine-tuning or classification head. Variate embeddings are randomly initialised for each downstream dataset. Output tokens are mean-pooled into a single global representation per sample. For each evaluation episode, $k = 5$ training samples per class are randomly drawn as the support set; their representations are averaged into one prototype per class. Test samples are then classified by cosine similarity to these prototypes, i.e. assigned to the nearest class prototype in the encoder's representation space. To reduce variance from the random choice of support samples, this prototypical $k$-shot procedure is repeated over 5 independent episodes with freshly drawn support sets; balanced accuracy is reported as the mean $\pm$ standard deviation. Random seeds are fixed so that all three general-purpose encoders receive identical support sets in each episode.

Table 14: Summary of baselines used in our experiments, including details on architecture and pre-training strategy. CL, MDM, and GPT refer to contrastive learning, masked data modelling, and autoregressive pre-training, respectively. GP denotes general-purpose encoder.

| Task | Method | Architecture | Pre-training | |
|------|--------|--------------|--------------|--|
| | | | **Strategy** | **Dataset (Time points)** |
| Classification | TS2Vec 2022 | 1D-CNN | CL | SleepEEG* (74 M) 2000 |
| | TimesNet 2022 | 2D-CNN | – | – |
| | TF-C 2022 | Transformer | CL | SleepEEG* (74 M) 2000 |
| | Ti-MAE 2023 | Transformer | MDM | SleepEEG* (74 M) 2000 |
| | SimMTM 2023 | Transformer | MDM | SleepEEG* (74 M) 2000 |
| | PatchTST 2023 | Transformer | – | – |
| | TSLANet 2024 | Transformer | – | – |
| | CroSSL 2024 | 1D-CNN | – | – |
| | InvConvNet 2025 | 1D-CNN | – | – |
| | Handcrafted 2022 | Statistical | – | – |
| | ShallowNet 2017 | 2D-CNN | – | – |
| | DeepNet 2017 | 2D-CNN | – | – |
| | EEGNet 2018 | 2D-CNN | – | – |
| | CBraMod 2025 | Transformer | MDM | TUEG (127 B) 2016 |
| | CSBrain 2025 | Transformer | MDM | TUEG (127 B) 2016 |
| Regression | PatchTST 2023 | Transformer | – | – |
| | ECG-Transformer 2021 | Transformer | – | – |
| | TSLANet 2024 | Transformer | – | – |
| | CroSSL 2024 | 1D-CNN | – | – |
| | InvConvNet 2025 | 1D-CNN | – | – |
| | ECG-MAE 2022 | Transformer | MDM | UK BioBank (2.4 B) 2015 |
| | CM-AE 2023 | 1D-CNN | MDM and CL | UK BioBank (2.4 B) 2015 |
| | MMCL 2025 | Transformer | MDM and CL | UK BioBank (2.4 B) 2015 |
| Forecasting | N-BEATS 2019 | Non-Linear Model | – | – |
| | DLinear 2023 | Linear Model | – | – |
| | PatchTST 2023 | Transformer | – | – |
| | GPT4TS 2023 | Transformer | GPT | ‡ |
| | TTM 2024 | Transformer | MDM | Custom (1 B) |
| | MOIRAI 2024 | Transformer | MDM | Custom (27 B) |
| | Chronos 2024 | Transformer | GPT | Custom (84 B) |
| | Timer-XL 2025 | Transformer | GPT | Custom (56 B) |
| | Time-MoE 2025 | Transformer | GPT | Custom (309 B) |
| | ROSE | Transformer | MDM | – |
| | DTAF | Transformer | MDM and KL | – |
| GP | UniTS 2024 | Transformer | MDM | Custom (36 B) |
| | MOMENT 2024 | Transformer | MDM | Custom (1.2 B) |

\* $371,055$ uni-variate, 2-seconds EEG recordings sampled at a frequency of 100 Hz.
‡ Method uses a GPT2 2018 encoder pre-trained on 10 billion text tokens.

Table 15: **Effect of NCC loss weight $\lambda$ on 5-shot prototypical classification performance. Best and second-best** balanced ACC (in % ↑) highlighted per dataset.

| Dataset | | | | | Performance @ NCC $\lambda$ | | |
|---------|---|---|---|---|---|---|---|
| **Domain** | **Name** | **Variates** | **Time points** | **Classes** | $\lambda\!=\!0.0$ | $\lambda\!=\!0.1$ | $\lambda\!=\!0.2$ |
| Electromechanics | Machine fault | 1 | 5120 | 3 | 73.7 ±6.4 | **75.4** ±5.8 | 72.8 ±3.4 |
| Acceleration | Hand gesture | 3 | 206 | 8 | 54.7 ±3.6 | 57.5 ±1.9 | **60.5** ±2.6 |
| EMG | Muscular disease | 1 | 1500 | 3 | 93.9 ±3.9 | **95.9** ±0.7 | 92.8 ±4.4 |
| | Epilepsy | 1 | 178 | 2 | 80.3 ±5.1 | 85.8 ±2.8 | **87.4** ±2.6 |
| EEG | Sleep stage | 1 | 3000 | 5 | **47.9** ±5.6 | 47.1 ±5.0 | 45.3 ±5.7 |
| | Event type | 19 | 1000 | 6 | 39.4 ±5.2 | **40.5** ±6.5 | 39.5 ±6.2 |
| | **Average** | | | | 65.0 ±5.0 | **67.0** ±3.8 | 66.4 ±4.2 |

**Results.** Tables 16 and 17 report prototypical 5-shot classification per dataset for the standard benchmark and the UEA multi-variate archive, respectively, and Table 18 summarises the results across all datasets

Table 16: **Prototypical 5-shot classification. Best** and second best balanced accuracy (bACC in % ↑) highlighted. Each encoder is frozen after pre-training; output tokens are mean-pooled, and $k=5$ randomly drawn training samples per class are averaged into class prototypes. Test samples are classified by cosine similarity to these prototypes, averaged over 5 runs with independently drawn training samples. For all datasets, $V$, $T$, and $N$ denote the number of variates, the number of time points, and the number of classes, respectively. MOMENT (386 M) achieves the highest average 5-shot accuracy but its memory and latency scale steeply with $T$ (up to 145 MB and 369 ms), making it impractical for deployment. UniTS (8.2 M) scales best with input size $V$ and $T$ but trails in performance. In contrast, OTIS (7.1 M) is wearable-feasible ($T \leq 1500$, $V \leq 3$: <10 MB, <4 ms), while ranking second in 5-shot performance and dominating full fine-tuning.

| Dataset | Encoder | Full FT | 5-Shot bACC | Memory [MB/Sample] ↓ | Energy [kWh/1M Samples] ↓ | Latency [ms/Sample] ↓ | Throughput [Samples/s] ↑ |
|---|---|---|---|---|---|---|---|
| **Machine fault** ($V=1, T=5120, N=3$) | MOMENT-386M | 92.8 ±2.4 | **69.0** ±7.2 | 92.4 | 25.59 | 369.01 | 2.7 |
| | UniTS-8.2M | 74.9 ±5.2 | 61.4 ±2.7 | 4.2 | 0.18 | 2.64 | 378.3 |
| | OTIS-7.1M | **98.1** ±0.7 | 62.6 ±4.9 | 9.9 | 0.27 | 3.84 | 260.3 |
| **Hand gesture** ($V=3, T=206, N=8$) | MOMENT-386M | 61.8 ±1.7 | **53.3** ±4.0 | 26.3 | 1.64 | 23.70 | 42.2 |
| | UniTS-8.2M | 61.5 ±0.9 | 40.8 ±4.3 | 1.2 | 0.02 | 0.36 | 2780.4 |
| | OTIS-7.1M | **78.0** ±2.0 | 48.8 ±3.0 | 1.0 | 0.04 | 0.53 | 1872.8 |
| **Muscular disease** ($V=1, T=1500, N=3$) | MOMENT-386M | 97.6 ±0.0 | **94.6** ±1.0 | 29.0 | 1.00 | 14.11 | 70.9 |
| | UniTS-8.2M | 97.1 ±1.1 | 93.2 ±2.4 | 1.1 | 0.01 | 0.12 | 8522.6 |
| | OTIS-7.1M | **97.6** ±0.0 | 93.2 ±2.8 | 1.5 | 0.02 | 0.37 | 2686.7 |
| **Epileptic seizure** ($V=1, T=178, N=2$) | MOMENT-386M | 91.8 ±0.7 | **93.9** ±1.0 | 23.3 | 0.52 | 7.43 | 134.7 |
| | UniTS-8.2M | 87.8 ±1.2 | 86.4 ±1.6 | 0.6 | 0.01 | 0.10 | 9741.2 |
| | OTIS-7.1M | **91.8** ±1.4 | 92.1 ±1.1 | 0.7 | 0.02 | 0.26 | 3780.3 |
| **Sleep stage** ($V=1, T=3000, N=5$) | MOMENT-386M | **75.6** ±0.2 | 33.9 ±1.2 | 29.8 | 2.24 | 33.16 | 30.2 |
| | UniTS-8.2M | 71.0 ±1.1 | 31.4 ±2.5 | 2.3 | 0.02 | 0.24 | 4155.0 |
| | OTIS-7.1M | 74.9 ±0.6 | **44.1** ±8.7 | 4.0 | 0.03 | 0.42 | 2375.5 |
| **Event type** ($V=19, T=1000, N=6$) | MOMENT-386M | 51.7 ±4.4 | **53.3** ±3.0 | 144.8 | 10.31 | 93.06 | 10.7 |
| | UniTS-8.2M | 54.6 ±2.1 | 36.9 ±5.3 | 16.6 | 0.06 | 0.55 | 1808.1 |
| | OTIS-7.1M | **61.4** ±1.3 | 41.9 ±4.7 | 104.2 | 0.19 | 1.72 | 582.4 |
| **Average** | MOMENT-386M | 78.6 ±1.6 | **66.3** ±2.9 | 57.6 | 6.88 | 90.08 | 48.6 |
| | UniTS-8.2M | 74.5 ±1.9 | 58.4 ±3.1 | 4.3 | 0.05 | 0.67 | 4564.3 |
| | OTIS-7.1M | **83.6** ±1.0 | 63.8 ±4.2 | 20.2 | 0.10 | 1.19 | 1926.3 |

used in our study. Averaged across all datasets, OTIS (7.1 M) matches MOMENT (386 M) in accuracy (59.3 % vs 59.7 %) and clearly outperforms UniTS (8.2 M; 51.5 %). On deployment metrics, OTIS consumes 10× less memory (3.5 vs 34.5 MB/sample), 43× less energy (0.03 vs 1.32 kWh/1M samples), and 37× lower latency (0.50 vs 18.5 ms/sample) than MOMENT, at 8× higher throughput (3,574 vs 425 samples/s). UniTS is the most efficient (1.6 MB, 0.21 ms, 10,844 samples/s) but trails both encoders by 8 pp in accuracy. Overall, OTIS is the only encoder that achieves remarkable few-shot accuracy while remaining highly deployable, indicating that its learned features are both expressive and well-structured for high-quality prototype-based diagnostics.

Table 17: **Prototypical** 5-**shot classification on** 23 **UEA archive datasets. Best** and second best balanced accuracy (bACC in % ↑) highlighted. MOMENT (386 M) is powerful but unsuitable for deployment on wearables and edge devices (up to 531 MB and 353 ms). UniTS (8.2 M) scales best with $V$ and $T$ but trails in performance. OTIS (7.1 M) achieves the highest average 5-shot performance while remaining wearable-feasible.

| Dataset | Encoder | 5-Shot bACC | Wins | Memory [MB/Sample] ↓ | Energy [kWh/1M Samples] ↓ | Latency [ms/Sample] ↓ | Throughput [Samples/s] ↑ |
|---|---|---|---|---|---|---|---|
| EthanolConcentration ($V=3, T=1751, N=4$) | MOMENT-386M | 24.3 ±2.9 | | 60.4 | 19.7 | 283.9 | 3.5 |
| | UniTS-8.2M | 25.9 ±2.3 | | 4.7 | 0.1 | 1.6 | 644.1 |
| | OTIS-7.1M | **26.4** ±3.8 | | 10.1 | 0.4 | 5.4 | 184.5 |
| Handwriting ($V=3, T=152, N=26$) | MOMENT-386M | **18.5** ±0.2 | | 25.3 | 2.2 | 31.6 | 31.7 |
| | UniTS-8.2M | 10.7 ±0.5 | | 1.1 | 0.0 | 0.6 | 1571.8 |
| | OTIS-7.1M | 11.1 ±0.7 | | 0.9 | 0.0 | 0.6 | 1592.4 |
| JapaneseVowels ($V=12, T=26, N=9$) | MOMENT-386M | 41.2 ±3.4 | | 24.0 | 1.4 | 19.6 | 51.1 |
| | UniTS-8.2M | 38.8 ±5.4 | | 2.2 | 0.1 | 1.6 | 633.8 |
| | OTIS-7.1M | **61.3** ±3.7 | | 0.8 | 0.0 | 0.4 | 2611.0 |
| SelfRegSCP1 ($V=6, T=896, N=2$) | MOMENT-386M | 62.7 ±11.1 | | 61.3 | 19.5 | 279.0 | 3.6 |
| | UniTS-8.2M | 68.1 ±7.5 | | 5.1 | 0.1 | 1.8 | 551.7 |
| | OTIS-7.1M | **68.5** ±4.9 | | 10.6 | 0.4 | 5.5 | 181.4 |
| SelfRegSCP2 ($V=7, T=1152, N=2$) | MOMENT-386M | **52.8** ±4.1 | | 81.1 | 24.5 | 352.8 | 2.8 |
| | UniTS-8.2M | 52.4 ±2.6 | | 7.2 | 0.1 | 1.9 | 523.7 |
| | OTIS-7.1M | 50.0 ±4.4 | | 21.7 | 0.6 | 8.4 | 119.6 |
| ArticularyWordRecognition ($V=9, T=144, N=25$) | MOMENT-386M | 77.3 ±1.7 | | 31.5 | 1.6 | 22.6 | 44.2 |
| | UniTS-8.2M | 60.0 ±2.3 | | 2.5 | 0.0 | 0.2 | 4235.6 |
| | OTIS-7.1M | **79.0** ±0.9 | | 1.8 | 0.0 | 0.3 | 2992.2 |
| AtrialFibrillation ($V=2, T=640, N=3$) | MOMENT-386M | 13.3 ±0.0 | | 103.0 | 1.5 | 20.6 | 48.6 |
| | UniTS-8.2M | 6.7 ±0.0 | | 2.6 | 0.0 | 0.3 | 3158.6 |
| | OTIS-7.1M | **33.3** ±0.0 | | 3.6 | 0.1 | 1.0 | 959.3 |
| BasicMotions ($V=6, T=100, N=4$) | MOMENT-386M | 93.5 ±1.2 | | 39.3 | 0.3 | 4.8 | 207.2 |
| | UniTS-8.2M | 75.0 ±11.6 | | 1.8 | 0.0 | 0.1 | 7401.4 |
| | OTIS-7.1M | **100.0** ±0.0 | | 1.4 | 0.0 | 0.4 | 2582.6 |
| CharacterTrajectories ($V=3, T=180, N=20$) | MOMENT-386M | **79.5** ±2.2 | | 25.8 | 0.3 | 4.6 | 217.4 |
| | UniTS-8.2M | 69.8 ±2.2 | | 1.2 | 0.0 | 0.1 | 10540.1 |
| | OTIS-7.1M | 75.3 ±1.2 | | 0.9 | 0.0 | 0.2 | 4179.7 |
| Cricket ($V=6, T=1197, N=12$) | MOMENT-386M | 68.6 ±1.9 | | 74.4 | 4.6 | 67.9 | 14.7 |
| | UniTS-8.2M | 79.7 ±3.4 | | 6.4 | 0.0 | 0.5 | 2116.6 |
| | OTIS-7.1M | **86.7** ±1.9 | | 17.0 | 0.1 | 1.0 | 1003.5 |
| ERing ($V=4, T=65, N=6$) | MOMENT-386M | **90.4** ±0.0 | | 23.8 | 0.2 | 2.2 | 461.9 |
| | UniTS-8.2M | 75.2 ±0.0 | | 1.1 | 0.0 | 0.1 | 12271.6 |
| | OTIS-7.1M | 84.4 ±0.0 | | 0.7 | 0.0 | 0.2 | 4108.4 |
| Epilepsy ($V=3, T=206, N=4$) | MOMENT-386M | 79.5 ±4.3 | | 26.3 | 0.7 | 9.7 | 103.3 |
| | UniTS-8.2M | 74.2 ±5.2 | | 1.2 | 0.0 | 0.1 | 12281.8 |
| | OTIS-7.1M | **84.3** ±4.7 | | 1.0 | 0.0 | 0.3 | 3986.7 |
| FaceDetection ($V=144, T=62, N=2$) | MOMENT-386M | 49.8 ±1.0 | | 147.2 | 4.6 | 65.9 | 15.2 |
| | UniTS-8.2M | 49.5 ±0.7 | | 25.7 | 0.2 | 2.2 | 457.6 |
| | OTIS-7.1M | **50.1** ±0.9 | | 16.5 | 0.1 | 1.0 | 1003.6 |
| FingerMovements ($V=28, T=50, N=2$) | MOMENT-386M | 50.9 ±0.4 | | 31.8 | 0.8 | 11.8 | 84.7 |
| | UniTS-8.2M | **52.3** ±2.7 | | 5.3 | 0.0 | 0.4 | 2428.5 |
| | OTIS-7.1M | 52.0 ±5.0 | | 1.9 | 0.0 | 0.2 | 4083.3 |
| HandMovementDirection ($V=10, T=400, N=4$) | MOMENT-386M | 24.4 ±4.8 | | 51.3 | 2.6 | 37.5 | 26.7 |
| | UniTS-8.2M | **26.9** ±3.7 | | 4.6 | 0.0 | 0.3 | 3092.5 |
| | OTIS-7.1M | 26.3 ±7.2 | | 6.5 | 0.0 | 0.5 | 1841.0 |
| Heartbeat ($V=61, T=405, N=2$) | MOMENT-386M | 54.7 ±7.3 | | 530.8 | 14.2 | 203.5 | 4.9 |
| | UniTS-8.2M | 57.1 ±1.8 | | 26.9 | 0.3 | 4.6 | 217.2 |
| | OTIS-7.1M | **61.0** ±2.6 | | 161.8 | 0.3 | 4.8 | 206.6 |
| LSST ($V=6, T=36, N=14$) | MOMENT-386M | **29.5** ±1.5 | | 23.3 | 0.1 | 1.7 | 577.2 |
| | UniTS-8.2M | 18.8 ±1.2 | | 1.4 | 0.0 | 0.1 | 11051.6 |
| | OTIS-7.1M | 26.4 ±3.2 | | 0.7 | 0.0 | 0.3 | 3973.6 |
| Libras ($V=2, T=45, N=15$) | MOMENT-386M | 45.1 ±2.4 | | 22.5 | 0.1 | 0.8 | 1276.7 |
| | UniTS-8.2M | 42.7 ±3.5 | | 0.7 | 0.0 | 0.1 | 15857.9 |
| | OTIS-7.1M | **46.9** ±2.4 | | 0.6 | 0.0 | 0.3 | 3905.7 |
| NATOPS ($V=24, T=51, N=6$) | MOMENT-386M | 50.2 ±4.6 | | 30.4 | 0.7 | 9.4 | 106.0 |
| | UniTS-8.2M | 33.9 ±3.2 | | 4.6 | 0.0 | 0.4 | 2823.1 |
| | OTIS-7.1M | **66.2** ±1.5 | | 1.7 | 0.0 | 0.2 | 4068.1 |
| Phoneme ($V=1, T=1024, N=39$) | MOMENT-386M | 13.0 ±0.4 | | 29.4 | 0.7 | 9.6 | 103.7 |
| | UniTS-8.2M | 12.4 ±0.9 | | 1.2 | 0.0 | 0.1 | 11930.2 |
| | OTIS-7.1M | **16.3** ±0.4 | | 1.6 | 0.0 | 0.3 | 2924.1 |
| RacketSports ($V=6, T=30, N=4$) | MOMENT-386M | 50.4 ±5.6 | | 23.0 | 0.2 | 2.7 | 372.2 |
| | UniTS-8.2M | 40.2 ±2.3 | | 1.3 | 0.0 | 0.2 | 5171.2 |
| | OTIS-7.1M | **57.3** ±3.2 | | 0.7 | 0.0 | 0.2 | 4151.4 |
| SpokenArabicDigits ($V=13, T=93, N=10$) | MOMENT-386M | 53.6 ±2.7 | | 29.5 | 1.3 | 18.4 | 54.4 |
| | UniTS-8.2M | 43.5 ±3.9 | | 3.0 | 0.0 | 0.2 | 4544.1 |
| | OTIS-7.1M | **55.0** ±3.8 | | 1.4 | 0.0 | 0.2 | 4071.1 |
| UWaveGestureLibrary ($V=3, T=315, N=8$) | MOMENT-386M | 60.6 ±1.0 | | 28.8 | 1.2 | 17.3 | 57.9 |
| | UniTS-8.2M | 52.1 ±2.0 | | 1.5 | 0.0 | 0.2 | 4942.6 |
| | OTIS-7.1M | **63.4** ±2.5 | | 1.4 | 0.0 | 0.3 | 3907.3 |
| Average | MOMENT-386M | 51.5 ±2.8 | 5 | 66.3 | 4.5 | 64.3 | 168.2 |
| | UniTS-8.2M | 46.3 ±3.0 | 2 | 4.9 | 0.1 | 0.8 | 5149.9 |
| | OTIS-7.1M | **55.7** ±2.6 | **16** | 11.5 | 0.1 | 1.4 | 2549.4 |

Table 18: **Average prototypical 5-shot classification across the standard, UEA, and UCR datasets.** **Best** and second best balanced accuracy (bACC in % ↑) per benchmark are highlighted; wins and deployment metrics are reported alongside. Each encoder is frozen after pre-training; output tokens are mean-pooled, and $k=5$ randomly drawn training samples per class are averaged into class prototypes; test samples are classified by cosine similarity to these prototypes, averaged over 5 runs with independently drawn training samples. The **Overall** row aggregates accuracy and per-sample deployment metrics over all 149 unique datasets (one UEA dataset overlaps with the standard benchmarks).

| Benchmark | Encoder | 5-Shot bACC | Wins | Memory [MB/Sample] ↓ | Energy [kWh/1M Samples] ↓ | Latency [ms/Sample] ↓ | Throughput [Samples/s] ↑ |
|---|---|---|---|---|---|---|---|
| **Standard** (6 datasets) | MOMENT-386M | **66.3** $\pm2.9$ | **5** | 57.6 | 6.88 | 90.08 | 48.6 |
| | UniTS-8.2M | 58.4 $\pm3.1$ | 0 | 4.3 | 0.05 | 0.67 | 4564.3 |
| | OTIS-7.1M | 63.8 $\pm4.2$ | 1 | 20.2 | 0.10 | 1.19 | 1926.3 |
| **UEA** (23 datasets) | MOMENT-386M | 51.5 $\pm2.8$ | 5 | 66.3 | 4.5 | 64.3 | 168.2 |
| | UniTS-8.2M | 46.3 $\pm3.0$ | 2 | 4.9 | 0.1 | 0.8 | 5149.9 |
| | OTIS-7.1M | **55.7** $\pm2.6$ | **16** | 11.5 | 0.1 | 1.4 | 2549.4 |
| **UCR** (121 datasets) | MOMENT-386M | **60.9** $\pm3.3$ | **63** | 27.3 | 0.437 | 6.3 | 492.2 |
| | UniTS-8.2M | 52.2 $\pm3.5$ | 11 | 0.9 | 0.005 | 0.1 | 12238.2 |
| | OTIS-7.1M | 59.8 $\pm3.4$ | 47 | 1.1 | 0.014 | 0.3 | 3850.6 |
| **Overall** (149 datasets) | MOMENT-386M | **59.7** $\pm3.2$ | **73** | 34.5 | 1.32 | 18.50 | 425.0 |
| | UniTS-8.2M | 51.5 $\pm3.4$ | 13 | 1.6 | 0.02 | 0.21 | 10844.0 |
| | OTIS-7.1M | 59.3 $\pm3.3$ | 64 | 3.5 | 0.03 | 0.50 | 3574.0 |

## H    Additional Results: Forecasting Visualisation

We visualise the generative capabilities of our encoder on eight forecasting datasets in Figure 9.

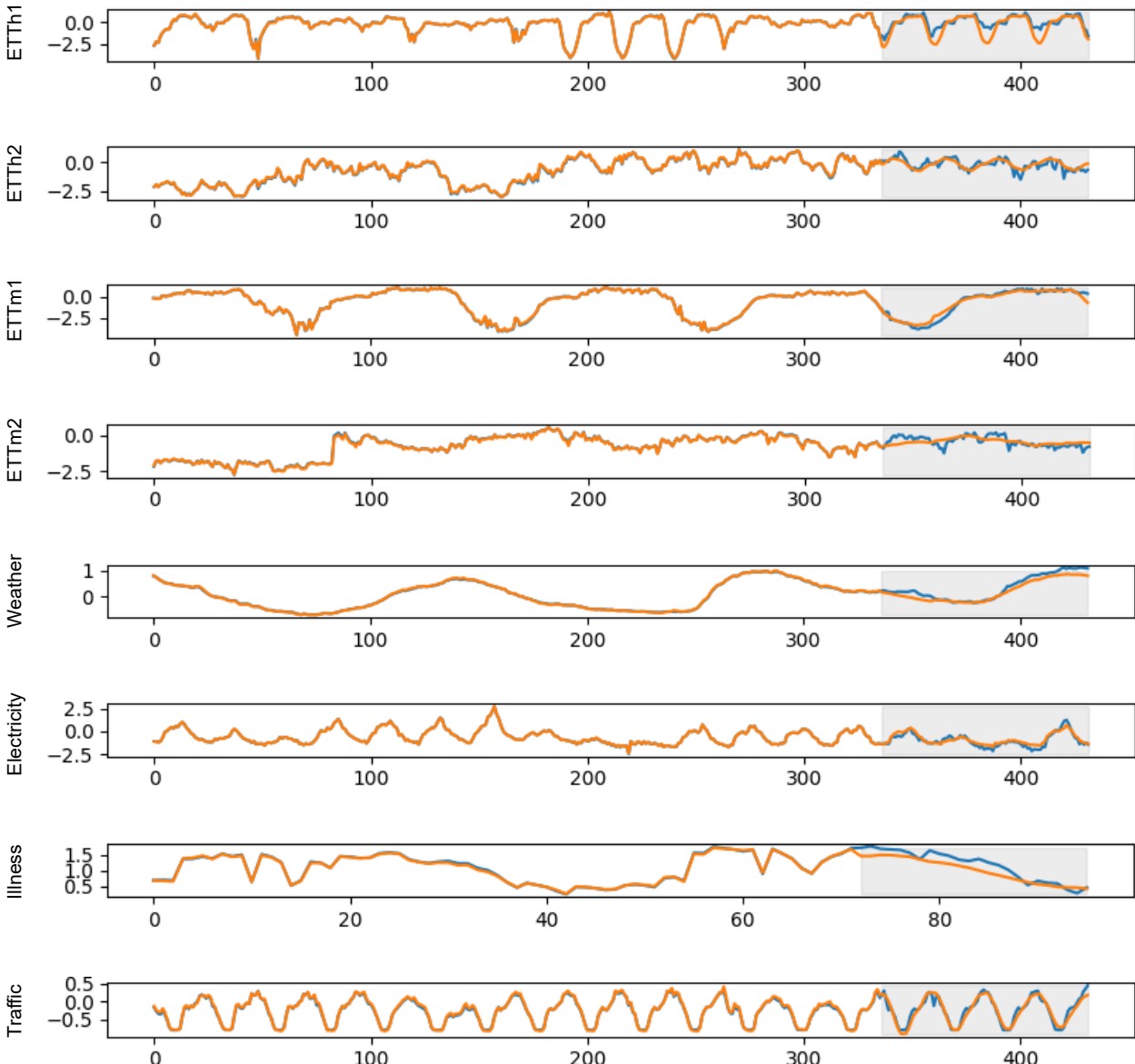

Figure 9: Visualisation of forecasting predictions. Ground truth in blue, prediction in orange. Areas highlighted in grey are not visible to the encoder. OTIS achieves high forecasting accuracy despite the tiny decoder (1.5 M) used for the predictions.

