# OpenReview forum: "OTIS: Learning High-Quality Time Series Features With Tiny Encoders"
_TMLR — Under review for TMLR_

### Review · Reviewer_ge2E · 2026-06-06

**Summary Of Contributions:**

The paper presents OTIS, a compact general-purpose time-series encoder designed to work across heterogeneous domains while remaining efficient enough for practical deployment. The main ideas are a domain-aware tokenizer, a dual masking strategy that combines reconstruction and causal prediction, and an MSE+NCC objective to improve structural reconstruction.

**Audience:**

Yes

**Audience Explanation:**

relevant to researchers working on time-series representation learning, self-supervised learning, efficient foundation models, and deployable ML.

**Claims And Evidence:**

Yes

**Claims Explanation:**

The paper evaluates OTIS on a broad set of tasks, compares it to much larger models, and includes ablations showing that the proposed tokenizer, masking strategy, and NCC loss each contribute to performance. The efficiency results also support the claim that the model is more deployable than larger alternatives.

That said, a few claims would be stronger with clearer experimental details. The authors should better document leakage checks, baseline tuning protocols, system-measurement settings, and medical split procedures. These do not undermine the core contribution, but they are important for reproducibility and for fully trusting the reported comparisons.

**Requested Changes:**

- Clarify how domain IDs and variate embeddings are assigned for unseen domains and datasets with different channel layouts.
- Provide explicit leakage-control details between pretraining data and downstream benchmarks.
- Describe the system-metric setup in detail, including hardware, precision, batch size, sequence length, and energy/latency measurement protocol.
- Specify the NCC implementation more clearly, including whether it is computed only on masked tokens and how numerical stability is handled.
- Clarify medical evaluation protocols, especially subject-wise or patient-wise splits.

Would strengthen the paper:

- Add sensitivity analysis for the random/post-fix masking ratio and the fixed post-fix horizon.
- Add a frozen linear-probe evaluation in addition to prototypical few-shot classification.
- Include robustness tests under noise, missingness, resampling, or distribution shift.
- Provide more details on baseline tuning to ensure fair comparisons.
- Expand related work discussion around channel/domain embeddings, masked forecasting, and structure-aware reconstruction losses.

---

> ### Author Response · Authors · 2026-07-02
>
> We thank Reviewer ***ge2E*** for recognising the breadth of our evaluation, the convincing ablations, and the efficiency results, and for the constructive reproducibility requests.
> ___
> ***C1) Domain IDs and variate embeddings for unseen domains and new variate layouts.*** Thanks for pointing this out. For inference on unseen domains, as well as datasets with different variate layouts, randomly initialised variate embeddings are utilised. This enables deployment on any time series from any domain while ensuring competitive downstream performance, as revealed by the few-shot experiments in Section 5 and Appendix G. Fine-tuning on just a single domain-specific sample already aligns them with the pre-trained space (see our reply to Reviewer VqPg). We have revised the methods section to clarify this.
> ___
> ***C2) Leakage, system metrics, baseline tuning, and robustness.*** While we are not big fans of cross-referencing and would love to provide you a detailed response, the character limit does not allow this. We therefore summarise these four points here and refer to our replies to Reviewer e4HC (W6 & C3, W5 & C4, W4 & C2, and C9) for the full detail. In brief: there is no overlap between pre-training and downstream data (Appendix A vs Appendix D; the FD-A/FD-B pair [1] differs strictly in operating conditions); deployment is measured on a single NVIDIA RTX A6000-48GB GPU under bfloat16 with a batch size of 1024, 10 warm-up and 25 timed passes, reporting per-sample metrics; all baselines are run from their official codebases with matching budgets and identical splits and sequence lengths; and robustness targets environmental deployability (noise, missing channels, distribution shift) rather than hardware deployability (memory, energy, latency), which we leave to dedicated future benchmarks, as noted in the Broader Impact Statement.
> ___
> ***C3) NCC implementation.*** The NCC loss is applied to all tokens, i.e. both visible and masked tokens, as defined in Equation 5. This global application ensures the preservation of the overall structural geometry, not just localised masked regions. Numerical stability is handled by adding a small epsilon to the variance. We have revised the methods section to clarify this.
> ___
> ***C4) Medical evaluation protocol.*** For all medical benchmarks, our study design ensures a clean subject-wise split: each subject, and all of their associated recordings, is exclusively assigned to either the training, validation, or test set. We have revised the experiments section to clarify this.
> ___
> ***C5) Masking sensitivity.*** Thanks for the suggestion. We conducted new experiments ablating (i) the post-fix probability, i.e. the ratio of post-fix to random masking (0.0, 0.25 = default, 0.5), and (ii) the post-fix masking ratio, i.e. the forecasting horizon (0.25, 0.5 = default, 0.75). We report the mean and standard deviation of online classification (UCR) and online forecasting (sine waves) across three pre-training seeds. First, using random masking alone (post-fix probability 0.0) hurts both tasks the most, with balanced accuracy dropping from 59.2 to 57.7 and forecasting MSE from 1.79 to 3.57, confirming that post-fix masking is essential. Increasing the post-fix probability to 0.5 is marginally better on classification (59.24 to 59.45) but worse on forecasting (1.79 to 2.15). Second, a long horizon of 0.75 degrades every metric (balanced accuracy 57.56, MSE 3.10). In short, the method is robust within a moderate band while both extremes degrade well beyond seed noise, supporting our defaults of a post-fix probability of 0.25 and a post-fix masking ratio of 0.5. We have updated the Appendix with these ablations.
> ___
> ***C6) k-NN and linear-probe evaluation.*** In addition to prototypical few-shot classification, we have now added weighted k-NN and linear probing on the UEA and UCR datasets for UniTS, MOMENT, and OTIS. We observe two consistent trends. First, the relative ranking is invariant to the evaluation protocol: OTIS > MOMENT > UniTS on the multivariate UEA archive, and MOMENT > OTIS > UniTS on the univariate UCR archive. Second, performance strictly follows a Linear (OTIS UEA: 68.1%) > k-NN (60.8%) > 5-Shot (55.7%) hierarchy across all models. This is expected: the prototypical evaluation is restricted to a highly constrained 5-shot setting, whereas both k-NN and linear probing leverage the entire training split. Between the two, linear probing achieves the highest performance by explicitly optimising an L2-regularised decision boundary with cross-validated hyperparameter tuning, whereas k-NN relies on a non-parametric weighted voting scheme with fixed hyperparameters. We have revised the results section to include these experiments.
> ___
> [1] Lessmeier, Christian, et al. "Condition monitoring of bearing damage in electromechanical drive systems by using motor current signals of electric motors: A benchmark data set for data-driven classification." PHM society European conference. 2016.

---

### Review · Reviewer_e4HC · 2026-06-09

**Summary Of Contributions:**

This paper introduces OTIS, a compact 7.1M-parameter time-series encoder designed to learn transferable representations for classification, regression, and short-horizon forecasting while remaining suitable for resource-constrained deployment. The method combines three main components: a domain-aware tokenizer with domain-specific variate embeddings, a dual masking strategy that mixes random and post-fix masking during pretraining, and an objective to better preserve global signal structure. Empirically, OTIS is evaluated across a wide range of tasks, including standard classification, EEG classification, ECG regression, few-shot classification on UEA/UCR datasets, and forecasting. The results suggest that OTIS achieves performance competitive with much larger models while requiring substantially less memory, energy, and latency, and ablation studies indicate that each proposed component contributes to the overall gains.

Strengths
- Strong deployment motivation. Compact and practical model. OTIS is only 7.1M parameters, yet is competitive with much larger models such as MOMENT and often stronger than UniTS on the tasks evaluated.
- Clear methodological components. Domain-aware tokenization, dual masking, and the NCC-augmented objective are simple, plausible, and well motivated for heterogeneous time-series data.
- Broad discriminative evaluation. The paper evaluates standard classification, EEG classification, ECG regression, UEA/UCR few-shot classification, and forecasting.
- Useful ablations. Removing the proposed components degrades performance, which supports the claim that the gains come from the proposed inductive biases rather than only from model size or tuning.
- Feature-quality evidence. The frozen few-shot experiments and representation visualizations support the claim that OTIS learns useful, structured embeddings.
- Efficiency reporting. Memory, energy, latency, and throughput comparisons are a valuable addition and align with the paper’s deployment-centered thesis.

Weaknesses
- Novelty over UniTS needs sharper positioning. UniTS already proposed a small, pre-trained, heterogeneous, multi-task time-series Transformer. OTIS’ novelty is better described as a new inductive-bias recipe for compact feature learning, not as the first compact general-purpose time-series model.
- Task scope is narrower than UniTS. OTIS does not evaluate imputation or anomaly detection, while UniTS does. This matters because the paper uses broad “general-purpose” language.
- OTIS is not a unified task-prompted model. UniTS supports multiple task types through task tokenization in one framework; OTIS uses an encoder plus downstream heads, and forecasting requires adding the decoder.
- Claims of state-of-the-art performance are too broad. OTIS is often competitive or best, but not uniformly best across all tasks. MOMENT remains stronger in some few-shot settings, and specialized forecasting models outperform OTIS on several forecasting datasets.
Baseline fairness is not fully clear. The paper should explain whether UniTS and other baselines were tuned with comparable budgets, preprocessing, sequence lengths, and validation protocols.
- Efficiency methodology needs more detail. Since deployment efficiency is central, the paper should specify measurement protocols for latency, memory, energy, batch size, precision, warm-up, and hardware.
- Possible pretraining/evaluation overlap should be clarified. The pretraining corpus includes related domains and datasets, so the paper should explicitly document leakage checks and dataset separation.
- Forecasting evidence is limited. OTIS’ forecasting results are useful but mixed, and they do not establish it as a general forecasting foundation model.
- Unseen-domain handling is underexplained. The domain-aware tokenizer relies on domain-specific variate embeddings; the paper should clarify how these are initialized/adapted for truly new domains.

**Audience:**

Yes

**Audience Explanation:**

Yes. The paper should be of interest to at least part of the TMLR audience, especially researchers working on time-series representation learning, self-supervised learning, foundation models, efficient Transformers, medical/physiological signal modeling, and edge deployment.

**Broader Impact Concerns:**

Broader impact statement in the paper is sufficient.

**Claims And Evidence:**

Yes

**Claims Explanation:**

The main claims are largely supported, especially the claim that OTIS learns useful time-series representations with a much smaller encoder than many existing general-purpose models. The paper provides good empirical evidence across classification, regression, few-shot transfer, and short-term forecasting, and the results generally show that OTIS is competitive with, and often better than, larger or similarly sized baselines. The evidence is particularly convincing for discriminative tasks: OTIS performs strongly on standard classification benchmarks, EEG tasks, ECG-based cardiac phenotype regression, and UEA/UCR few-shot classification.

The evidence for the deployment claim is directionally convincing. The paper reports much lower parameter count, memory, energy, and latency than MOMENT while maintaining similar average few-shot performance, which supports the argument that targeted inductive bias can improve deployability. This is one of the strongest aspects of the submission.

However, some of the paper’s broader claims are overstated relative to the evidence. The results do not show that OTIS is uniformly state of the art across all evaluated settings. MOMENT remains stronger in some few-shot cases, and specialized forecasting models outperform OTIS on several forecasting datasets. Therefore, claims such as “matches the state of the art” or “state-of-the-art performance” should be qualified more carefully. A more accurate statement would be that OTIS is often best or near-best among general-purpose encoders on the evaluated tasks while being substantially more efficient.

The evidence is also less complete for the claim that OTIS is a broadly “general-purpose” time-series model. Compared with UniTS, OTIS does not evaluate some important task families such as imputation and anomaly detection, and it is not demonstrated as a single unified task-prompted model across task types. The paper’s evidence is strongest for OTIS as a compact, transferable representation learner, rather than as a full replacement for all capabilities of prior unified time-series models.

Finally, the clarity and reproducibility of the evidence could be improved. Since the main claims depend heavily on comparisons to baselines and on efficiency measurements, the paper should provide more detail on baseline tuning, preprocessing, fine-tuning budgets, sequence lengths, energy/latency measurement protocols, and possible pretraining/evaluation overlap. Without these details, the results are promising and mostly convincing, but not yet fully conclusive.

Overall, I find the evidence generally credible, but I would ask the authors to temper the strongest claims and provide more methodological detail before treating the empirical comparisons and deployment claims as fully established.

**Requested Changes:**

Critical
- Clarify and temper the main claims. The paper should revise language such as “state-of-the-art performance” and “matches the state of the art” to reflect the actual mixed results. A more accurate claim would be that OTIS is often best or competitive with much larger general-purpose encoders while being substantially more efficient.
- Provide a clearer baseline-comparison protocol. The paper should specify, for each baseline group, whether models were reimplemented or run from official checkpoints, what preprocessing was used, how hyperparameters were tuned, what fine-tuning budgets were allowed, and whether all methods had comparable access to validation data and sequence lengths.
- Strengthen the leakage/overlap analysis. The paper should explicitly document that downstream evaluation datasets are not included in pretraining. For related datasets, such as FD-A/FD-B or datasets from the same repositories/domains, the authors should explain why the evaluation remains a fair test of generalization.
- Give full details for deployment metrics. Since efficiency is central to the contribution, the authors should provide a reproducible protocol for memory, energy, latency, and throughput measurements, including hardware, precision, batch size, number of runs, warm-up, software stack, and whether preprocessing/tokenization is included.

Non-critical
- Add statistical significance or confidence analysis for key comparisons. Many reported numbers are close, especially OTIS vs MOMENT in some few-shot and forecasting settings. Significance testing or paired comparisons across datasets would help establish when differences are meaningful.
- Expand the ablation study. The current leave-one-out ablations are helpful, but it would be useful to include interaction effects, e.g., domain-aware tokenization plus NCC without dual masking, or dual masking ratios other than 75/25.
- Better justify the domain-aware tokenizer for unseen domains. Since downstream tasks may come from domains not seen during pretraining, the paper should explain how new domain-specific variate embeddings are initialized and tuned, and how sensitive performance is to this choice.
- Separate representation-quality claims from forecasting claims. The forecasting results should be framed as an auxiliary demonstration rather than a primary SOTA claim. Additional long-horizon or zero-shot forecasting experiments would be needed to make stronger claims.
- Include more robustness evaluations. Since the paper motivates deployment in clinical and industrial settings, evaluations under noise, missing channels, sensor shifts, sampling-rate changes, and distribution shift would make the deployment story more convincing.
- Improve clarity around scaling experiments. The conclusion that performance saturates with model size is interesting, but it is based on a limited set of larger variants. The authors should clarify whether larger models were equally optimized and whether the result reflects architectural limits, optimization limits, or data limits.

---

> ### Author Response · Authors · 2026-07-02
>
> We thank Reviewer ***e4HC*** for recognising OTIS' deployment motivation, clear methodological components, broad evaluation, useful ablations, and feature-quality and efficiency evidence.
> ___
> ***W1 & W3) Positioning vs UniTS; encoder vs unified model.*** UniTS is a task-prompted model: a single 8.2M network with a built-in GEN tower and token that performs a task end-to-end. In contrast, OTIS is an encoder: it only reads the input and extracts features and, like vision encoders such as DINOv3, needs downstream heads for any task. Our novelty is an inductive-bias recipe for compact feature learning, not a claim of being the first compact general-purpose model; general-purpose here means the features transfer to any task and domain.
> ___
> ***W2 & W7 & C8) Generative task scope and forecasting framing.*** This is absolutely correct. Being an encoder, OTIS does not target imputation or anomaly detection, which UniTS handles through its GEN tower. Our short-term forecasting is only exploratory, showing the extracted features can also serve simple generative tasks once the lightweight 1.5M decoder is attached. We make **no state-of-the-art claim on forecasting** and have already framed generative experiments as exploratory.
> ___
> ***W4 & C2) Baseline-comparison protocol.*** All baselines, summarised in Appendix E (Table 14), are evaluated using their official codebases. Full fine-tuning follows the original works with matching budgets, while frozen few-shot uses the official checkpoints. All methods have identical access to validation splits and identical sequence lengths. We have reworked Appendix E to clarify this.
> ___
> ***W5 & C4) Deployment-metric protocol.*** Thanks for pointing this out. All metrics are measured on a single NVIDIA RTX A6000-48GB under bfloat16 autocast, with a fixed batch size of 1024 at each dataset's native sequence length T and variate count V. We discard 10 warm-up passes and average 25 timed iterations, timing the encoder forward including tokenisation but excluding data loading, normalisation, and transfer, via time.perf_counter, torch.cuda, and NVML (pynvml). All four metrics are per-sample and thus batch-size invariant. We have revised Appendix C to detail this.
> ___
> ***W6 & C3) Leakage and overlap.*** We appreciate the question. There is no overlap between pre-training (Appendix A, Table 6) and downstream data (Appendix D, Table 13). The only related pair is FD-A [1] (0.6% of the pre-training corpus) and FD-B (downstream): both are bearing-vibration signals from the same rig, but under strictly different working conditions (speed, load, radial force), so FD-B remains a genuine generalisation test. We have revised the results section to emphasise this.
> ___
> ***W8 & C7) Unseen-domain tokenisation.*** For truly new domains, randomly initialised variate embeddings are utilised, already yielding competitive frozen few-shot performance (Section 5, Appendix G). Fine-tuning on just a single domain-specific sample suffices to align a random embedding with the pre-trained space (see our reply to Reviewer VqPg). We have reworked the methods section to clarify this.
> ___
> ***C5) Key comparisons.*** We appreciate this. Our study aims to show tiny general-purpose encoders such as OTIS are *as competitive as* encoders like MOMENT. To this end, the study design including an extensive evaluation on (i) 162 datasets, (ii) 3 distinct task types, and (iii) established measures (mean, standard deviation, and wins) ensures sufficient testing. We are happy to discuss further if you strongly believe extending it adds new insights.
> ___
> ***C6) Expanded ablation.*** This is a great suggestion. Therefore, we added new ablations on the post-fix horizon and post-fix-to-random ratio, as detailed in our reply to Reviewer ge2E (R5). In brief: random-only masking hurts most and a long post-fix horizon degrades every metric, supporting our defaults.
> ___
> ***C9) Robustness.*** We agree robustness matters, but this targets *environmental* deployability (noise, missing channels, shift) warranting a dedicated study, as noted in the Broader Impact Statement. In contrast, our study scope is *hardware* deployability (memory, energy, latency).
> ___
> ***C10) Scaling.*** Very good point. The Large and Huge variants are optimised identically to the base OTIS. Interestingly, recent work shows the same: Moirai 2.0 [2] trains on 295B time points (versus 11B in our study) yet finds its smallest 11.4M variant to beat their larger 87M and 305M ones, suggesting the critical bottleneck is data quality and redundancy, not optimisation or data quantity (see our reply to Reviewer VqPg).
> ___
> [1] Lessmeier, Christian, et al. "Condition monitoring of bearing damage in electromechanical drive systems by using motor current signals of electric motors: A benchmark data set for data-driven classification." PHM society European conference. 2016.
>
> [2] Liu, Chenghao, et al. "Moirai 2.0: When less is more for time series forecasting." arXiv. 2026.

---

### Review · Reviewer_VqPg · 2026-06-22

**Summary Of Contributions:**

This paper introduces OTIS, a lightweight general-purpose time series encoder designed to address the deployment challenges of large models on resource-constrained edge devices. The authors challenge the scaling law in time series modeling, arguing that careful architectural inductive biases can replace brute-force parameter scaling. The authors propose three key components:
1. A domain-aware tokeniser that resolves semantic heterogeneity by injecting learnable domain-specific embeddings.
2. A dual masking strategy during pre-training to capture both spatiotemporal structures and temporal causality.
3. A structure-aware objective utilizing a Normalized Cross-Correlation loss combined with MSE, preventing mean-collapse and preserving structural fidelity.
Experiments across 162 downstream tasks show that the 7.1M OTIS matches or outperforms state-of-the-art large models like 386M MOMENT, while consuming significantly less memory, energy, and latency.

**Additional Comments:**

NA

**Audience:**

Yes

**Audience Explanation:**

Small and competitive models are valuable for deployment in real-world applications.

**Claims And Evidence:**

Yes

**Claims Explanation:**

1. The evaluation is conducted on 162 datasets, covering a wide range of tasks including classification, regression, and forecasting, spanning domains from healthcare to electromechanics.
2. The tiny model performs comparably to or better than large models on multiple metrics like memory, latency, energy and accuracy.
3. The effectiveness of the proposed inductive biases is convincingly validated through ablation studies (Table 5), which demonstrate the performance gain of the domain-aware tokeniser, dual masking, and NCC loss.
4. The claim regarding the diminishing returns of scaling laws is supported by experiments using 40M and 116M variants of OTIS, which show performance saturation.

**Requested Changes:**

1. It's interesting to see that increasing the model size yields diminishing returns. However, the 11B-time-points pre-training corpus is relatively small compared to the trillions of tokens used in LLMs. Could the saturation be an artifact of data starvation rather than an inherent property of time series modalities? Adding a brief discussion on whether data volume (rather than just parameter count) is the true bottleneck for time series scaling laws would be better.
2. In Appendix G, the authors mention that variate embeddings are randomly initialised for each downstream dataset for unseen domains. It's better to move the relevant explanation to the main text and briefly discuss how much fine-tuning is required for a randomly initialized embedding to align with the pre-trained space.

---

> ### Author Response · Authors · 2026-07-02
>
> We thank Reviewer ***VqPg*** for recognising the breadth of our evaluation across 162 datasets, the competitiveness of the tiny 7.1M encoder, the ablation evidence for our inductive biases, and the scaling analysis.
> ___
> ***C1) Is scaling saturation an artifact of data volume?*** This is a very good point. Interestingly, similar observations are made in recent works, indicating that data *volume* alone might not be the bottleneck for time series scaling laws. For instance, the authors in [1] utilise 36M samples (versus 640k samples in our study) and 295B time points (versus 11B in our study) to train their model, yet surprisingly find their smallest 11.4M variant (versus 7.1M in our study) to achieve the best performance among all variants (87M and 305M), despite this large pre-training corpus. So rather than data volume, it might be the **data quality and redundancy** that are critical when scaling time series models. Hence, we hypothesise data cleansing/curation to be a promising direction for future studies. We have added this discussion to the manuscript.
> ___
> ***C2) Move the variate-embedding explanation to the main text; discuss fine-tuning needs.*** Thanks for pointing this out. We have moved the explanation regarding the initialisation of variate embeddings from Appendix G to the main text (Section 3.4) and added a brief discussion on the fine-tuning requirement, as summarised in the following. While domain-specific variate embeddings are utilised during pre-training to address multi-domain data heterogeneity, the few-shot experiments in Section 5 and Appendix G reveal that, for inference on unseen domains, **randomly initialised** variate embeddings are sufficient. Interestingly, we find that fine-tuning on **just a single domain-specific sample is sufficient** to transform a randomly initialised variate embedding (Fig. 5a) into a domain-specific one (Fig. 5b) that perfectly aligns with the pre-trained representation space, as indicated by the encoder's strong generalisation to unseen samples from the new domain (Fig. 5c).
> ___
> [1] Liu, Chenghao, et al. "Moirai 2.0: When less is more for time series forecasting." arXiv. 2026.